# Truncation of mutant huntingtin in knock-in mice demonstrates exon1 huntingtin is a key pathogenic form

Huiming Yang[1,2,3,4], Su Yang[2], Liang Jing[4,5], Luoxiu Huang[4], Luxiao Chen[6], Xianxian Zhao[2], Weili Yang[2], Yongcheng Pan[1,4], Peng Yin[2], Zhaohui S Qin [6], Shihua Li[2✉] & Xiao-Jiang Li [2✉]

Polyglutamine expansion in proteins can cause selective neurodegeneration, although the mechanisms are not fully understood. In Huntington's disease (HD), proteolytic processing generates toxic N-terminal huntingtin (HTT) fragments that preferentially kill striatal neurons. Here, using CRISPR/Cas9 to truncate full-length mutant HTT in HD140Q knock-in (KI) mice, we show that exon 1 HTT is stably present in the brain, regardless of truncation sites in full-length HTT. This N-terminal HTT leads to similar HD-like phenotypes and age-dependent HTT accumulation in the striatum in different KI mice. We find that exon 1 HTT is constantly generated but its selective accumulation in the striatum is associated with the age-dependent expression of striatum-enriched HspBP1, a chaperone inhibitory protein. Our findings suggest that tissue-specific chaperone function contributes to the selective neuropathology in HD, and highlight the therapeutic potential in blocking generation of exon 1 HTT.

[1] Department of Neurology, The First Affiliated Hospital, Sun Yat-sen University, 510080 Guangzhou, China. [2] Guangdong-Hongkong-Macau Institute of CNS Regeneration, Ministry of Education CNS Regeneration Collaborative Joint Laboratory, Jinan University, 510632 Guangzhou, China. [3] Department of Neurology, Xiangya Hospital, Central South University, 410008 Changsha, Hunan, China. [4] Department of Human Genetics, Emory University School of Medicine, Atlanta, GA 30322, USA. [5] Department of Emergency, Tongji Hospital, Tongji Medical College, Huazhong University of Science and Technology, 430030 Wuhan, China. [6] Department of Biostatistics and Bioinformatics, Rollins School of Public Health, Emory University, Atlanta, GA 30322, USA. ✉email: lishihualis@jnu.edu.cn; xjli33@jnu.edu.cn

Expanded polyglutamine (polyQ) repeats in different proteins cause Huntington's disease (HD) and at least eight other neurodegenerative diseases[1]. Although these disease proteins are widely expressed and differ in their structures and functions, the polyQ diseases share a common feature, which is age-dependent neurodegeneration in distinct brain regions[1]. In HD, the expanded polyQ domain (>37Q) in huntingtin (HTT) is encoded by the CAG repeats in exon 1 of the HTT gene and causes neurodegeneration that preferentially occurs in the striatum[2–4]. It remains unclear how polyQ expansion can result in selective neurodegeneration in each polyQ disease.

Strong evidence has shown that expanded polyQ causes proteins, especially cleaved protein fragments, to misfold and abnormally accumulate in the nucleus and nerve terminals[1,3,5]. In HD knock-in mice that endogenously express polyQ-expanded HTT, the nuclear accumulation and aggregation of mutant HTT preferentially occur in striatal neurons in brain[6–11], which is consistent with the most severe vulnerability of striatal neurons in the brains of HD patients. Biochemical studies provide further evidence that polyQ expansion causes N-terminal fragments of HTT to misfold and abnormally interact with a large number of proteins, leading to impairment of multiple cellular functions[2,3,12]. Consistently, the presence of various N-terminal fragments containing the expanded polyQ repeat is evident in the HD mouse brain that expresses full-length mutant HTT endogenously[13–16]. In support of the idea that N-terminal HTT fragments are toxic, transgenic mice expressing mutant exon 1 HTT and other small mutant N-terminal HTT fragments show much more severe and progressive phenotypes than mice expressing full-length mutant HTT[17,18].

As generation of various mutant N-terminal HTT in the brain is due to proteolysis of full-length HTT, considerable efforts have been devoted to identifying the critical cleavage sites in HTT that are responsible for neuropathology[19]. Despite extensive studies, it remains unknown which proteolytic cleavage site(s) are essential for producing toxic N-terminal HTT fragments. Moreover, aberrant exon 1–intron RNA in the HD brain was found to generate exon 1 HTT[20], further complicating the mechanism for generating toxic N-terminal HTT fragments. As targeting the HTT gene to lower its expression is a promising way to treat HD, understanding the nature of the pathological form of mutant HTT is imperative for determining the targeting site in the HTT gene.

Identifying the critical pathological form of mutant HTT needs rigorous investigation of mutant HTT that is expressed at the endogenous level. This is because HTT toxicity is largely dependent on the polyQ length and the expression level of mutant HTT. Overexpression of mutant HTT would confound the neuropathology and phenotypes. To this end, we used CRISPR/Cas9 to target the HTT gene at different sites in HD140Q knock-in mice, which could yield truncated N-terminal HTT fragments that carry the same polyQ repeat number and are expressed at the endogenous level. The newly generated HD KI mice, which express different N-terminal HTT fragments, allowed for rigorous comparison of the phenotypes and neuropathology caused by different forms of mutant HTT. Our findings revealed that an N-terminal mutant HTT fragment equivalent to exon 1 HTT is constantly generated, stable, and responsible for the nuclear HTT accumulation and aggregation in the striatum. Moreover, we found that HspBP1, an inhibitor of Hsp70 and co-chaperone CHIP, is abundantly expressed in the striatum in an age-dependent manner. Inhibiting HspBP1 in the striatum can efficiently reduce the nuclear accumulation and aggregation of mutant HTT in the striatum of HD KI mice. Given that considerable attention has been devoted to posttranslational modifications of multiple sites in the non-exon 1 regions of HTT, our findings suggest that gene therapeutics and drug treatment of HD should focus on the elimination of exon 1 HTT, a key pathological form of HTT that has a critical role in the selective neurodegeneration in HD.

## Results

**Deletion of exon 1 HTT is not deleterious to mouse**. To generate new HD KI mice that express different N-terminal mutant HTT fragments at the endogenous level, we used CRISPR/Cas9 to target the HTT gene in HD140Q knock-in mice. Guide RNAs (gRNAs) were designed to disrupt the HTT gene at different sites, which would result in the expression of truncated HTT (Fig. 1a). These gRNAs were co-injected with Cas9 into fertilized mouse embryos from wild mice (WT) or HD140Q KI mice. Transfer of these embryos into female recipient mice led to newborn founders (F0). These F0 mice were then mated with WT mice to produce F1 and F2 generations. Genotyping of F1 or F2 newborn mice and sequencing of the targeted regions of the HTT gene confirmed indel mutations that caused frame-shifts and generated truncated HTT (Supplementary Fig. 1). However, we were unable to identify any homozygous newborn mice that carry truncated HTT mutations at exon 2, 13, or 31 (Fig. 1b). These results indicate that truncated HTT is unable to support early embryonic development.

A mutation that eliminates 177 nucleotides (d177) encoding 59 amino acids including the polyQ and polyproline (polyP) domains without frameshift was able to reach homozygosity (Fig. 1c). Genotyping showed that this mutation reduces exon 1 HTT DNA size (Supplementary Fig. 2a), confirming the in-frame deletion in exon 1 HTT. RT-PCR using primers for exon 1 and 2 HTT showed that the mutations did not affect the expression level of mutant exon 1 HTT mRNA (Supplementary Fig. 2b). Western blotting of the homozygous d177 mice verified that HTT with exon 1 deletion was smaller than full-length HTT but was expressed at the same level and in the same distribution pattern as full-length HTT (Fig. 1d). Characterization of homozygous d177 mice revealed that they were indistinguishable from WT mice in development and motor function (Fig. 1e). Taken together, these results suggest that the truncated N-terminal HTT proteins are unable to support early embryonic development, but removing the majority of amino acids encoded by exon 1 can still retain the critical function of HTT.

**Generation of HD KI mice expressing N-terminal mutant HTT**. The above described new HD KI mouse models indicate that while exon 1 is not essential for early development, it may cause a gain of toxicity if carrying an expanded CAG repeats. To analyze the toxicity of truncated mutant HTT, we focused on the heterozygous KI mice that could survive but express truncated HTT proteins containing the expanded polyQ repeats. We were able to obtain KI mice with CRISPR/Cas9-mediated truncations at exon 2 and exon 13 in the mutant allele (Supplementary Fig. 3). We found that these mutations did not affect mRNA expression of the truncated HTT (Supplementary Fig. 4) and resulted in truncated HTT consisting of the first 96 (KI-96) or 571 (KI-571) amino acids with an expanded polyQ (140Q) repeat (Fig. 2a). The newly generated KI mice carrying these mutations were named KI-96 and KI-571 mice for further studies.

To characterize the expression of N-terminal HTT at the protein level in the new KI mice, we tried a panel of antibodies to HTT and found that only mouse EM48 (mEM48), which was previously generated by us and reacts with the VA residues after the polyP domain[21], could detect mutant HTT in the new KI mice (Supplementary Fig. 5). However, compared with HD140Q knock-in (KI-FL) mouse brain in which full-length mutant HTT

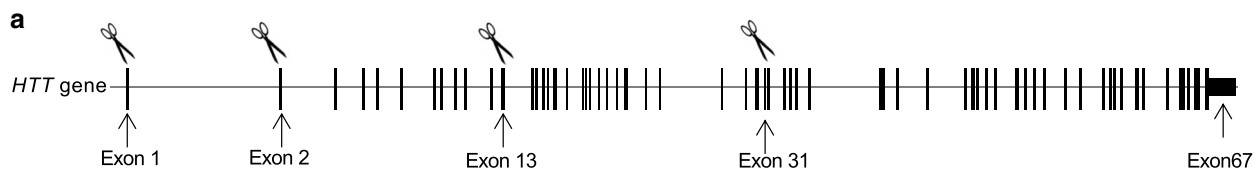

**a**

HTT gene

Exon 1    Exon 2    Exon 13    Exon 31    Exon67

**b**

The numbers and percentage of live birth mice of different homozygous genotypes

| Genotype | WT | Het | Hom | Hom % total |
|---|---|---|---|---|
| d177 | 13/52 | 24/52 | 15/52 | 28.8 |
| Exon 2-T | 8/21 | 13/21 | 0/21 | 0 |
| Exon13-T | 14/55 | 41/55 | 0/55 | 0 |
| Exon 31-T | 10/31 | 21/31 | 0/31 | 0 |

**c**

PAM    gRNA7 cutting site                              PAM    gRNA1 cutting site

WT    ATGGCAACCCTGGAAAAGCTGATGAAGGCTTT  CAG repeats  CTGCCAGGTCCGGCAGAGGAACCGCTGCACCGACC

d177  ATGGCA---------------------------------------------------------------GAGGAACCGCTGCACCGACC (–177)

Exon 1 HTT

WT    MATLEKLMKAFESLKSFQQQQQQQPPPQAPPPPPPPPPPQPPQPPPQGQPPPPPPPLPGPAEEPLHRPKKELSATKKDRVNHCL
d177  MA------------------------------------------------------------------EEPLHRPKKELSATKKDRVNHCL (–59)

Huntingtin                                          d177 (-Exon 1)

1                                                   3144

**d**

Wild type              d177/d177

Cortex  Cerebellum  Striatum  Hippocampus  Cortex  Cerebellum  Striatum  Hippocampus

WT HTT →                                          ← d177 HTT

250 kD –

37 kD –    GAPDH

125 kD –    Vinculin

**e**

Rotarod (s)
Balance beam (s)
Body weight (g)

WT    Hom d177    Het d177

4 m    6 m    8 m

was expressed and a series of N-terminal fragments were visible, only one HTT fragment around 50 kD was seen in the brain of KI-571 mice. It is likely that this N-terminal HTT fragment is stable in KI mice and possesses a unique conformation that can be readily recognized by mEM48. In support of this idea, immunohistochemical staining of the striatum of KI-571 and KI-FL mice with different antibodies (1C2 and mEM48) showed the comparable levels of 1C2 and mEM48 immunoreactive nuclear HTT and aggregates (Supplementary Fig. 6). It has been well documented that small N-terminal HTT fragments are able to

**Fig. 1 CRISPR/Cas9 targeting endogenous *HTT* at different sites in mice. a** Different *HTT* exons were targeted by CRISPR/Cas9 to generate truncated HTT. **b** The summary showing the survival rates of homozygous mice with exon 1 deletion (d177) or HTT truncations at different exons (Exon 2-T, 13-T, and 31-T). WT wild type, Het heterozygous, Hom homozygous. **c** Sequencing analysis confirmed that d177 mutant mice have 177 nucleotides deletion in exon 1, resulting in deletion of N-terminal 59 amino acids without changing other amino acid sequences in the mouse HTT. **d** Western blotting analysis of d177 mouse brains showing that the d177 HTT (-exon 1) is smaller than full-length mouse HTT (WT HTT). The blots were probed by the antibody MAB2166. More than three times of experiments were performed. **e** The growth and motor functions of heterozygous (Het) or homozygous (Hom) d177 and wild-type (WT) littermate mice. Data are presented as minimum to maximum showing all points. A *p*-value < 0.05 is considered as significance, and two-way ANOVA followed by Tukey's multiple comparison tests was used for analysis (mice numbers: $n = 8$ for WT, $n = 10$ for Hom d177, $n = 8$ for Het d177. Rotarod, $F = 0.07562$, $p = 0.9272$; balance beam, $F = 0.1509$, $p = 0.8602$; body weight, $F = 0.106$, $p = 0.8996$). Data are presented as mean values ± SEM. Source data are provided as a Source Data file.

accumulate in the nucleus to form aggregates[22,23]. Thus, the ~50 kD band seen on the western blots is likely present in both KI-571 and KI-FL mouse striatum to accumulate in the nucleus but its unique conformation only allows mEM48 to detect it via western blotting. The size of this HTT fragment is similar to exon 1 HTT that was found in the brains of different HD mice after immunoprecipitation of mutant HTT[14,20]. Furthermore, aggregated HTT in the stacking gel was initially seen in the striatal tissues, but not in the cortex and cerebellum, of KI-571 mouse, indicating the preferential HTT aggregation in the striatum (Fig. 2b).

Using brain tissues from KI-571 mice at different ages, we found that mutant HTT was aggregated in an age-dependent manner. In KI-571 mice at 6 months of age, aggregated HTT was predominantly seen in the striatum (Fig. 2c). When KI-571 mice became old (11 months), other brain regions (cortex and cerebellum) also showed aggregated HTT, though the striatum contained much more abundant aggregated HTT.

We also obtained KI-96 mice in which N-terminal HTT fragment equivalent to exon 1 HTT was generated. Western blotting of their brain tissues revealed the same expression pattern as KI-571 mice: the striatum showed aggregated HTT and an HTT fragment in the same size as the smallest HTT fragment in the KI-FL mouse brain (Fig. 2d). We also saw that mutant HTT in the striatum in different KI mice accumulated in the neuronal nuclei to the same extent and in an age-dependent manner (Fig. 2e). Thus, all KI mice, regardless of the expression of full-length or truncated N-terminal HTT fragments, have a stable N-terminal HTT fragment that becomes aggregated preferentially in the striatum.

**Preferential nuclear accumulation of exon 1 like HTT in the striatum.** Generation of HD mice expressing full-length mutant HTT via either knock-in or transgenic approaches leads to an important finding that mutant HTT preferentially accumulates in the striatum at the early disease stage[6–11]. Although such selectivity in HTT accumulation remarkably mirrors the preferential loss of striatal neuronal cells in the brains of HD patients[2–4], it remains unknown which N-terminal HTT fragment is able to preferentially accumulate in the striatum. The western blotting results also show that mutant HTT in KI-571 and KI-96 mice is equivalent in size to exon 1 HTT and is able to preferentially accumulate in the striatum (Fig. 2b–e). Comparison of the distribution of mutant HTT in these different KI mouse brains revealed the similar age-dependent and selective accumulation of mutant HTT in neuronal nuclei in the striatum (Fig. 3a, Supplementary Fig. 7). Quantification of nuclear HTT staining also indicates that mutant HTT is selectively accumulated in the striatum in all KI mice examined (Fig. 3b). Given that the stable N-terminal HTT in all KI mice has similar polyQ repeats and the protein size is comparable to exon 1 HTT in R6/2 mice, we concluded that exon 1 HTT or its equivalent product is stable and

can preferentially accumulate in the striatum in all HD KI mice generated in our study.

However, transgenic exon 1 HTT in R6/2 mice is broadly expressed in different brain regions[22]. It is possible that overexpression of mutant exon 1 HTT under an exogenous promoter may lose the selectivity for mutant exon 1 HTT to accumulate in the striatum. Indeed, R6/2 mice express much more abundant HTT aggregates at 8 weeks of age when compared with different KI mice at 6 months of age (Fig. 4a, b). Using PCR to amplify the *HTT* DNA containing the CAG repeat, we confirmed that the size of the CAG repeats is similar in R6/2 and all KI mice (Fig. 4c), suggesting that difference in the abundance of HTT aggregates in these HD mice is not due to the repeat length difference but is related to the expression level of mutant HTT. Consistently, western blotting showed that the size of transgenic exon 1 HTT in R6/2 mouse was similar to that of the stable HTT fragments in all KI mouse striatum and that the transgenic HTT was at the higher level and formed much more aggregated HTT in the striatum (Fig. 4d). RT-PCR indeed revealed that the exon 1 *HTT* mRNA level was more abundant in R6/2 mouse striatum than in KI mouse striatum (Fig. 4e, f). Quantification of the ratio of HTT mRNA to actin mRNA verified the higher level of mutant exon 1 *HTT* in R6/2 mouse striatum than in KI mouse striatum (Fig. 4g), which was also confirmed by quantitative RT-PCR (qPCR) (Fig. 4h). Thus, R6/2 mice express a much higher level of exon 1 *HTT* so that they show very severe and progressive phenotypes[22].

Exon 1 HTT fragment was found to be generated by aberrant exon 1-intron 1 RNA, which was observed only in the HD brains in which a large CAG repeat is present[20,24]. Using RT-PCR of WT and KI mouse, we confirmed the selective expression of aberrant exon 1-intron 1 RNA in the KI mouse striatum when compared with WT mouse striatum (Supplementary Fig. 8a, b). By examining RNAseq results, we also found that aberrant exon 1-intron 1 RNA was present in the KI mouse striatum, but its level appeared to be lower than the normally spliced exon 1–exon 3 mRNA (Supplementary Fig. 8c). To more rigorously compare the levels of canonical HTT mRNA (exon 1–exon 3) and aberrantly spliced mRNA (exon 1–intron 1), we performed RT-PCR to detect their levels in the same PCR reactions by including a critical control without reverse transcriptase to ensure that PCR products seen in the mouse brains are specifically derived from cDNAs, rather than genomic DNAs. Using this assay, we found that canonical HTT exon 1–exon 3 mRNA is more abundant than the aberrant exon 1–intron 1 RNA in heterozygous KI mice (Supplementary Fig. 8d, e). Using real time PCR to quantify the relative levels of exon 1–exon 3 and exon 1–intron 1, we verified that the aberrant spliced HTT mRNA (exon 1–intron 1) is specifically present in the KI mouse striatum but its level is lower than normally spliced mutant HTT (exon 1–exon 3) (Supplementary Fig. 8 f-g). The results suggest that, although the aberrant exon 1–intron 1 RNA can produce mutant exon 1 HTT, this small N-terminal HTT fragment may be predominantly

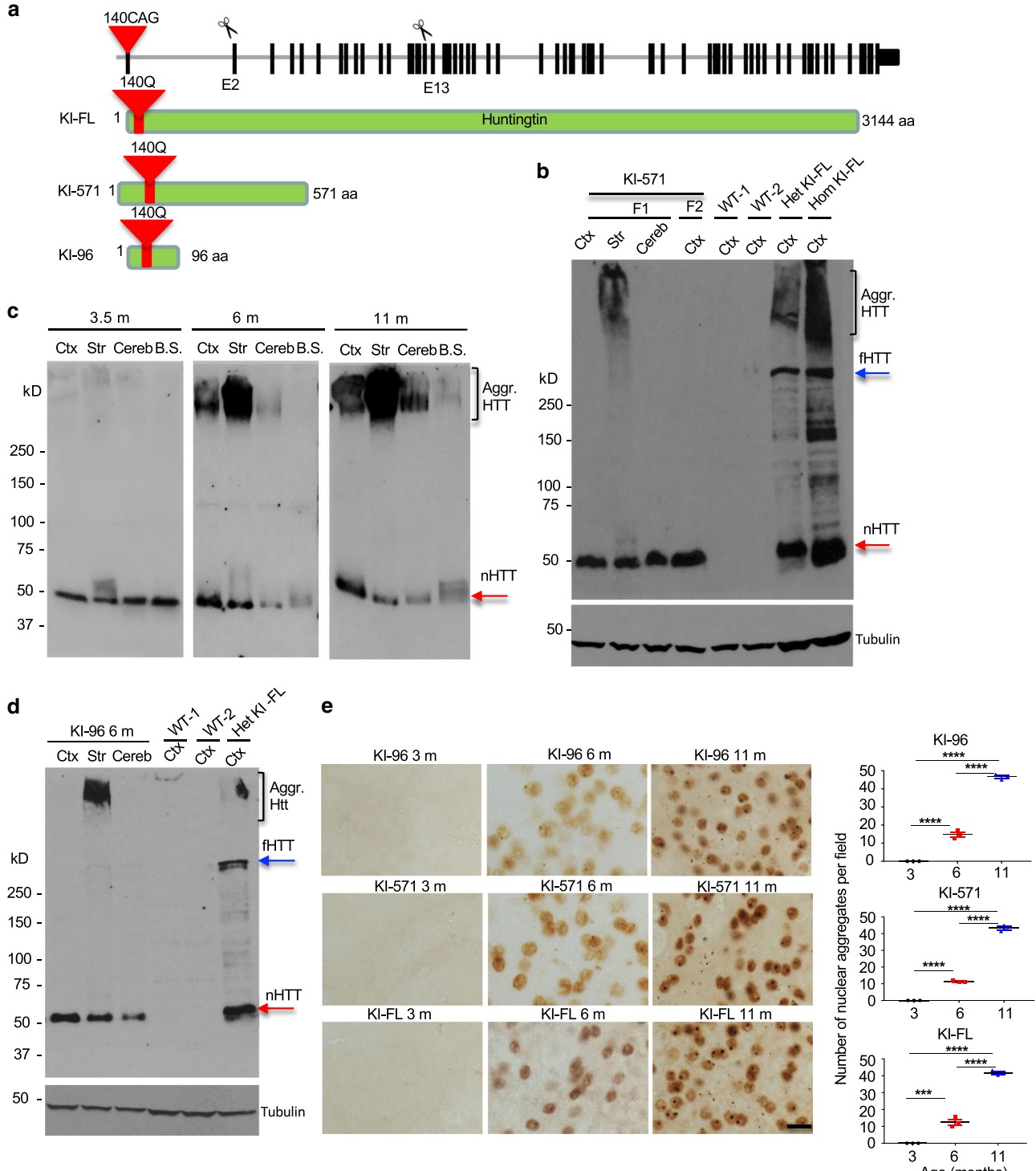

**Fig. 2 Generation of HD KI mice expressing different N-terminal HTT fragments. a** Generation of new HD KI mouse models by truncating exon 2 and exon 13 of the mouse *HTT* gene in HD140Q KI (KI-FL) mice via CRISPR/Cas9, resulting in the expression of truncated mutant HTT containing the first 96 and 571 amino acids in KI-96 and KI-571 mice, respectively, with an additional 140Q repeat. **b** mEM48 western blotting showing the specific expression of mutant HTT in KI-571 mouse brains. This N-terminal mutant HTT (nHTT) is at the same size as the smallest mutant HTT in KI-FL mouse brain and forms aggregated HTT in the striatum. **c** mEM48 western blotting revealing an age-dependent aggregation of mutant HTT in KI-571 mouse brain. **d** mEM48 western blotting showing the specific expression of mutant HTT in KI-96 mice, which also has the same size as the smallest mutant HTT in KI-FL mouse brain and preferentially forms aggregated HTT in the striatum. More than three times of experiments were performed independently for **b–d**. In **a–d**, Ctx cortex, Str striatum, Cereb cerebellum. F1 F1 generation, F2 F2 generation. Aggr. HTT aggregated HTT, nHTT n-terminal HTT, fHTT full-length HTT. **e** mEM48 immunostaining showing an age-dependent accumulation of mutant HTT in the striatum in KI-96, KI-571, and KI-FL mice. Scale bar: 10 μm. Quantification of immunostaining results in Fig. 2e were obtained by counting six images per brain region per mouse (*n* = 3 mice per genotype, ***p < 0.001, ****p < 0.0001, one-way ANOVA followed by Tukey's multiple comparison tests, KI-FL, F = 408.1, p < 0.0001; KI-571, F = 1195, p < 0.0001; KI-96, F = 720.8, p < 0.0001). Data are presented as mean values ± SEM. Source data are provided as a Source Data file.

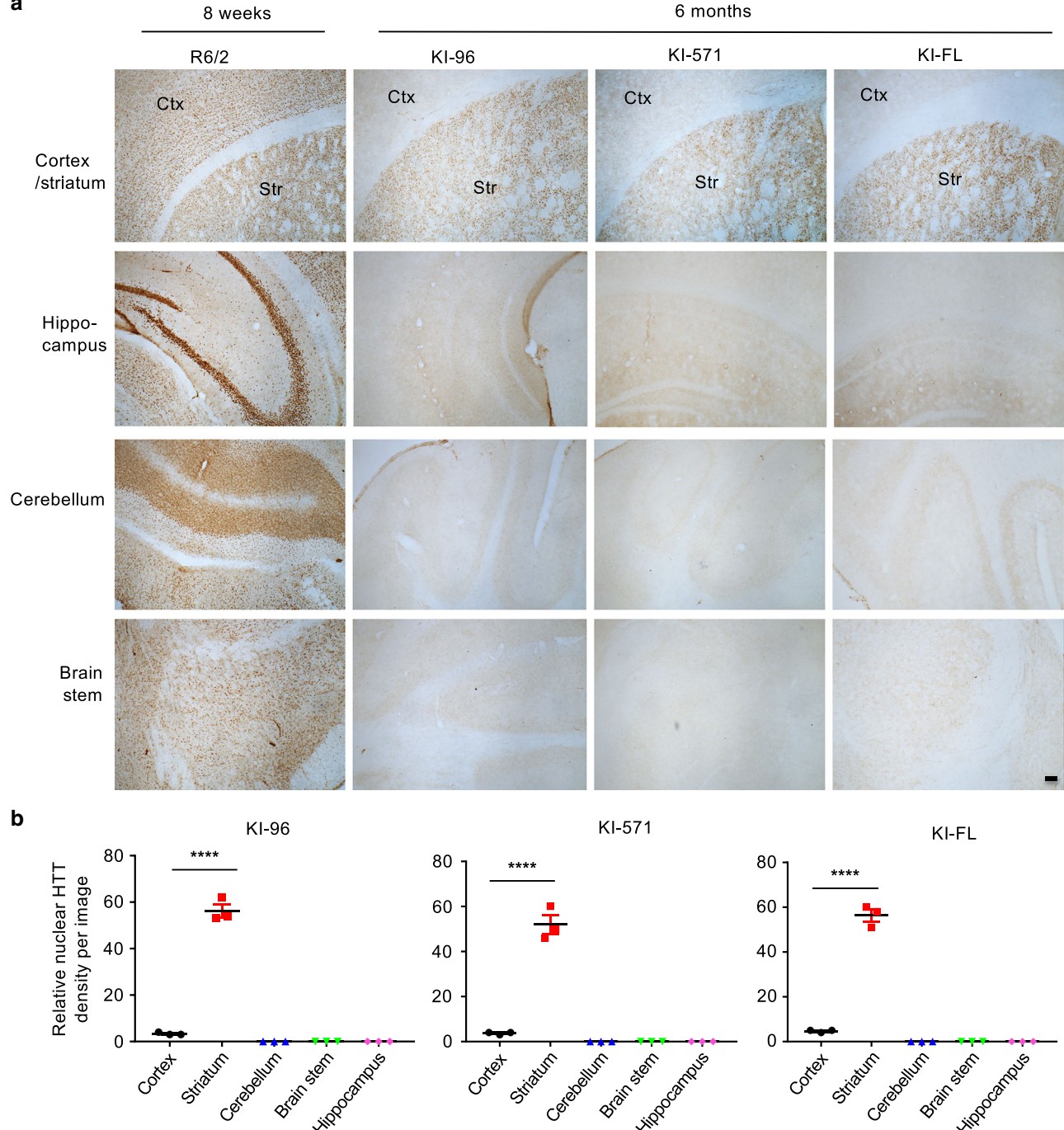

**Fig. 3 Preferential accumulation and aggregation of mutant HTT in the striatum of different KI mice. a** Comparison of mEM48 staining of various brain regions in R6/2 and KI (KI-96, KI-571, KI-FL) mice showing the preferential accumulation of mutant HTT in the striatum of KI mice. The micrographs (5×) from the striatum (Str), cortex (Ctx), hippocampus, cerebellum, and brain stem are presented. Scale bar: 100 μm. **b** Quantitative analysis of density of mEM48 labeled nuclei in different brain regions in KI mice. The data were obtained by counting six images per brain region per mouse ($n = 3$ mice per genotype, ****$p < 0.0001$, one-way ANOVA followed by Tukey's multiple comparison tests, KI-FL, $F = 405.5$, $p < 0.0001$; KI-571, $F = 150.3$, $p < 0.0001$; KI-96, $F = 375.9$, $p < 0.0001$). Data are presented as mean values ± SEM. Source data are provided as a Source Data file.

generated from proteolytic cleavage of full-length or a longer HTT protein fragments.

**Promotion of HTT cleavage does not increase the nuclear accumulation of mutant HTT.** Examining HD FL-KI mice at different ages revealed that more aggregated HTT and less soluble mutant exon 1 HTT were present in the striatum of older KI mice

(Fig. 5a). This finding suggests that exon 1 HTT may be constantly generated by proteolytic cleavages and/or aberrant splicing in the striatum but its nuclear accumulation and aggregation are age-dependent. The similar nuclear accumulation and aggregation of mutant HTT in different KI mouse brains also suggest that additional truncation of mutant HTT did not significantly facilitate the nuclear accumulation and aggregation of mutant HTT. To further test this hypothesis, we used stereotaxic injection of

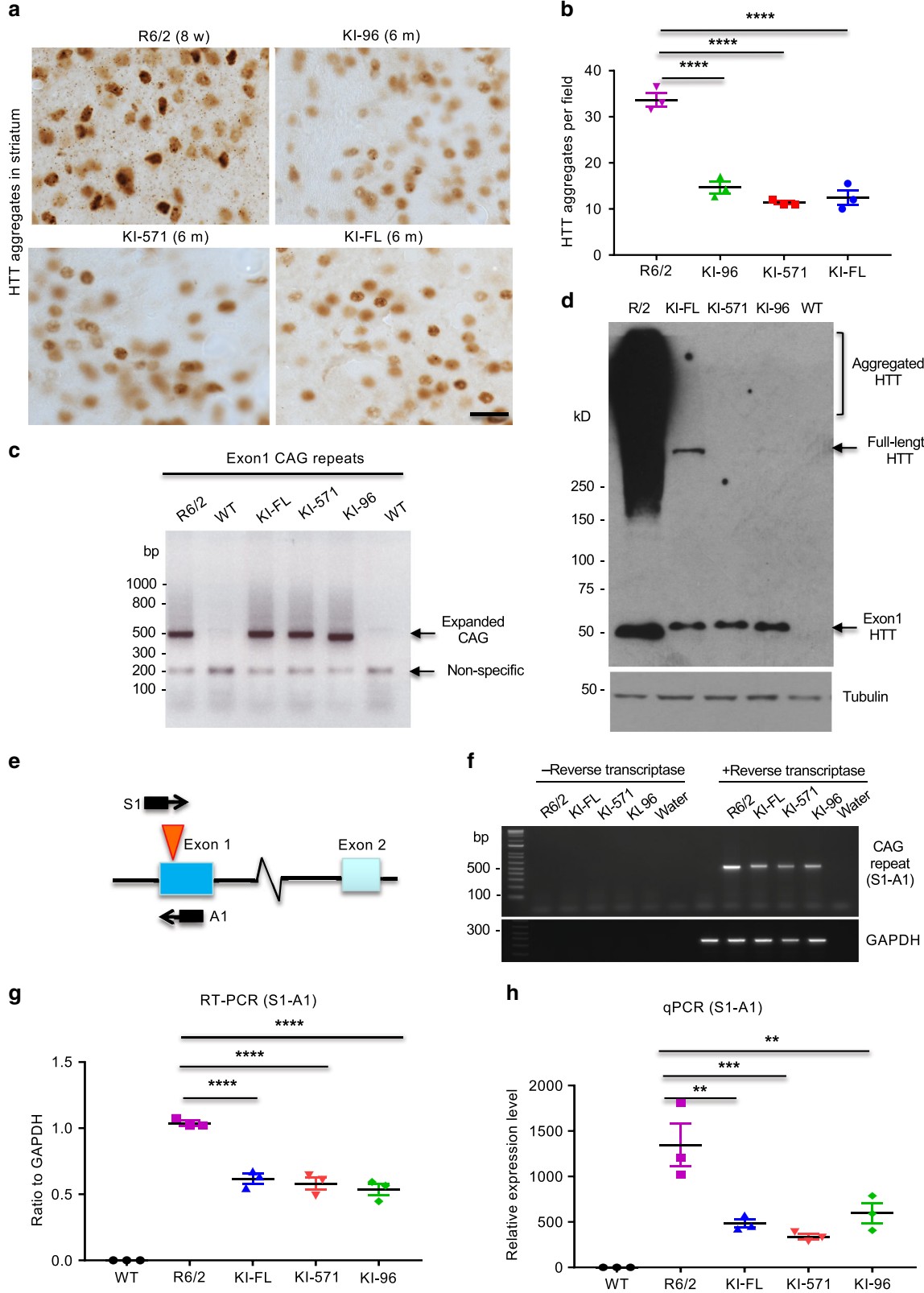

AAV CRISPR/Cas9 to truncate the *HTT* gene at the region corresponding to amino acid 91 and 571 in adult KI-FL mouse brain. These N-terminal HTT fragments are similar to exon 1 HTT and caspase-cleaved products[25,26], respectively, which have been extensively investigated for their roles in HD pathogenesis[27–30]. For this experiment, we obtained mice that express Cas9 under the pCAG promoter after Cre-recombination[31]. These mice were

crossed with EIIa-Cre transgenic mice to generate Cas9 mice that ubiquitously express Cas9 (Supplementary Fig. 9a). We then crossed Cas9 mice with HD140Q KI (KI-FL) mice to generate KI/Cas9 mice so that these mice can be injected with only AAV HTT-gRNA to truncate full-length mutant HTT (Fig. 5b). PCR and western blotting assays confirmed the expression of Cas9 in the striatum of KI/Cas9 mice (Supplementary Fig. 9b, c). Using

**Fig. 4 Comparison of the expression levels of mutant HTT in HD KI mouse striatum. a** Immunostaining of the mouse striatum with mEM48 showing that R6/2 mice have much more abundant nuclear HTT staining than KI-96, KI-571, and KI-FL mice. Scale bar: 20 μm. **b** Quantitative analysis of the density of nuclear HTT aggregates in the striatum of R6/2 and different KI mice. The data were generated by mean value of counting six images each mouse ($n = 3$ mice per genotype, ****$p < 0.0001$, one-way ANOVA followed by Tukey's multiple comparison tests, $F = 69.11$, $p < 0.0001$). Data are presented as mean values ± SEM. **c** PCR to amplify the CAG repeat showing R6/2 and all KI mice carry the similar CAG repeat numbers. WT wild type. **d** Western blotting using mEM48 revealing that R6/2 mouse striatum expresses more abundant soluble mutant HTT and aggregated HTT than KI mouse striatum. R6/2 mouse at 8 weeks and KI mice at 3.5 months of age were examined. More than three independent experiments were performed. **e** PCR primers used to detect mutant *HTT* transcripts in R6/2 and HD KI mouse brains. **f** RT-PCR revealing a much higher level of exon 1 *HTT* mRNAs in R6/2 mice than other KI mice. The negative controls are PCR results from mRNA samples without reverse transcriptase to rule out genomic DNA amplification products. **g** The ratios of *HTT* cDNA bands to the control gapdh band in **f**. Data are presented as mean values ± SEM and were obtained from three independent experiments each mouse ($n = 3$ mice per genotype, ****$p < 0.0001$, one-way ANOVA followed by Tukey's multiple comparison tests, $F = 114.9$, adjusted $p < 0.0001$). **h** Quantitative RT-PCR using primers S1 and A1 also confirmed the significantly higher level of exon 1 *HTT* mRNA in the R6/2 mouse striatum than KI mouse striatum. Data are presented as mean values ± SEM and were obtained from three independent experiments each mouse ($n = 3$ mice per genotype, **$p < 0.01$, ***$p < 0.001$, one-way ANOVA followed by Tukey's multiple comparison tests, $F = 17.44$, $p = 0.0002$). Source data are provided as a Source Data file.

homozygous HD KI/Cas9 mice at the age of 2 months, we performed stereotaxic injection of AAV HTT-gRNA, which targets HTT exon 2 or exon 13 to generate N-terminal HTT consisting of the first 91 or 571 amino acids. The control was the same AAV vector expressing a control gRNA that does not target to any mouse gene. The AAV gRNA vector also expresses RFP under the CMV promoter such that the injected striatum shows a broad distribution of RFP (Fig. 5c), indicating an efficient AAV transduction. Two months after injection, T7E1 digestion of the targeted HTT gene in the AAV-injected brain region showed the selective targeting of *HTT* gene by HTT-gRNA, but not control gRNA (Fig. 5d). Although western blotting revealed that HTT-gRNA-mediated truncation could decrease the level of full-length HTT and increase the amount of soluble exon 1 HTT, the amount of HTT aggregates shown in the stacking gel remained similar in the control gRNA- and HTT-gRNA-injected striatal regions (Fig. 5e). Also, immunohistochemical staining demonstrated no obvious difference in the nuclear HTT staining between the control gRNA- and HTT-RNA-injected striatal regions (Fig. 5f). Thus, the nuclear accumulation of mutant HTT does not seem to be influenced by altering cleavage of mutant HTT in the non-exon 1 regions, which is consistent with similar nuclear HTT accumulation and aggregation in the striatum of different KI (KI-96, KI-571, and KI-FL) mice.

**Phenotypes of KI mice expressing different N-terminal HTT fragments.** Next, we wanted to investigate whether truncation of full-length mutant HTT at amino acids 96 and 571 can promote neurological phenotypes, based on the fact that smaller N-terminal HTT fragments are more prone to misfolding. We examined motor functions of heterozygous HD KI mice to compare their phenotypes because homozygous KI-96 and KI-571 are not viable and the motor deficits in HD KI mice have been reliably replicated in many previous studies[9,16,32,33]. All KI (KI-96, KI-571, and KI-FL) mice showed significantly defective motor functions starting from 7 months when compared with WT mice. Reduced rotarod performance was seen at 8 months of age whereas grip strength defect occurred at 7 months of age (Fig. 6a). Consistently, poor balance beam performance of all KI mice appeared at 9 months of age. Thus, among these KI mouse models, the poor motor performances appeared to be similar, though KI-96 mice displayed more severe balance beam deficits. All these KI mice developed normally with similar body weight and do not appear to be different from age-matched WT mice before the age of 11 months.

Although heterozygous HD140Q KI mice showed mild neuropathologic changes, HTT aggregates and gliosis (increased Gfap) in HD KI mice have been repeatedly reported by different

groups[8,11,32,34–36]. However, we did not observe different Gfap staining signals between KI-571 and KI-FL mice, though they showed more reactive astrocytes than WT mice (Fig. 6b, c). The undetectable differences between KI-571 and KI-FL mice suggests that mutant exon 1 HTT, which is stable in both mouse models, elicits similar neuropathology in the brain.

It is well documented that gene transcriptional dysregulation is more severe in the striatum of HD KI mice, consistent with the abundant nuclear accumulation of HTT in the striatum[37]. We thus performed RNAseq to compare gene expression profiling in the striatum of KI-96, KI-571, and KI-FL mice. Using transcriptional modules described previously for dysregulated gene expression in HD KI mice[37], we found that all KI mice showed similar alterations in these modules (Supplementary Fig. 10). Thus, by comparing behavior, pathology, and gene expression in KI-96, KI-571, and KI-FL mice, we found these KI mice display similar defects, regardless of the expression of different N-terminal mutant HTT fragments. Taken together, the results support the idea that the stable and common exon 1 HTT fragment causes similar phenotypes in different types of KI mice.

**HspBP1 accounts for the preferential accumulation of mutant HTT in the striatum.** By examining the expression of mutant HTT in HD140Q KI mice from 2 to 9 months of age, we found that mutant exon 1 HTT is constantly produced in the cortex and striatum in young and old mice but is selectively aggregated in the striatum in aged mice (Fig. 7a). This phenomenon suggests that a striatal specific factor may contribute to the preferential accumulation and aggregation of mutant HTT in striatal neurons. We also noticed that mutant HTT is preferentially accumulated in neuronal cells but not glial cells. We previously identified that HspBP1, an inhibitor of Hsp70 and co-chaperone CHIP[38–40], is abundantly expressed in neuronal cells but not in glial cells and can promote HTT misfolding by inhibiting chaperone function[41]. To investigate whether HspBP1 is involved in the selective nuclear accumulation of mutant HTT, we first examined HspBP1 expression in the striatum, cortex, and cerebellum in WT mice at different ages. Interestingly, HspBP1 is more abundant in the striatum than the cortex and cerebellum and its expression in the striatum is increased with age (Fig. 7b). The selective enrichment of HspBP1 in the striatum was also demonstrated by HspBP1 immunohistochemical staining of the mouse brain (Supplementary Fig. 11). However, RT-PCR revealed similar levels of HspBP1 mRNA in different mouse brain regions (Supplementary Fig. 12), suggesting that selectively high level of HspBP1 in the striatum is regulated at the protein level.

Next, we wanted to investigate whether knocking down HspBP1 in the KI-FL striatum can alter the nuclear accumulation

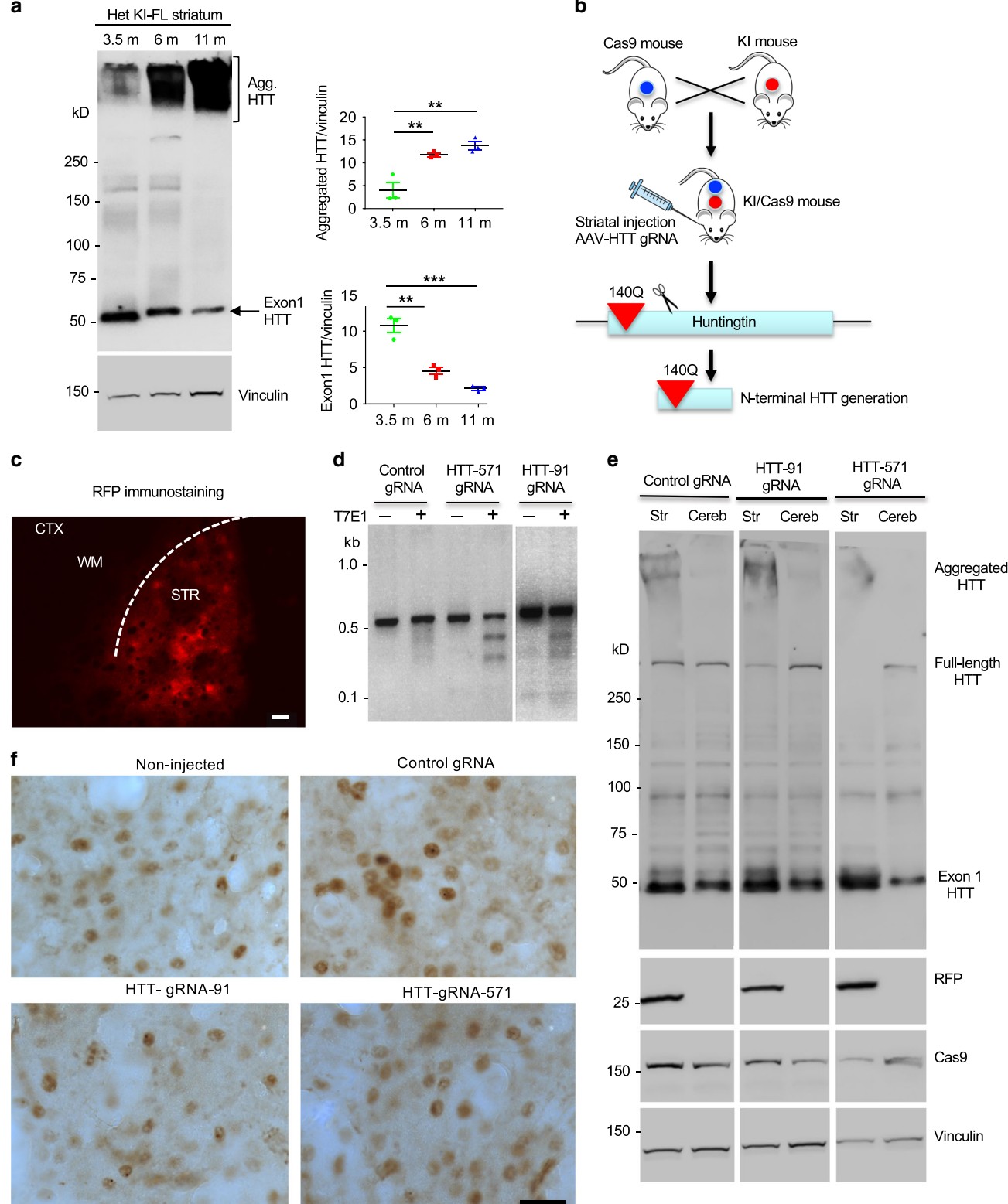

of mutant HTT. To this end, we used gRNA to target the endogenous mouse HspBP1 and verified that expression of HspBP1-gRNA via AAV could effectively reduce HspBP1 in cultured mouse striatal neuronal cells (Fig. 7c). We then performed stereotaxic injection of AAV HspBP1-gRNA and AAV control gRNA in the right and left striatum, respectively, of HD KI/Cas9 mice at age of 25 weeks (Fig. 7d) in order to rigorously compare nuclear mutant HTT staining in the same

mouse brain. The results revealed that knocking down HspBP1 could significantly reduce nuclear accumulation of mutant HTT (Fig. 7e). Quantification of the density of nuclear HTT verified that elimination of HspBP1 could significantly diminish the nuclear accumulation of mutant HTT in the striatum (Fig. 7f). Given that mutant exon 1 HTT is prone to misfold and that chaperones are important for reducing protein misfolding, the higher level of HspBP1 in the striatal neurons may yield stronger

**Fig. 5 Truncation of mutant HTT in the striatum of adult HD140Q KI mice. a** mEM48 western blotting showing age-dependent decrease of soluble mutant exon 1 HTT and corresponding increase of aggregated HTT in the HD KI striatum. Quantification of western blotting results is also shown. Data are presented as mean values ± SEM and were obtained from three independent experiments each mouse ($n = 3$ mice per genotype, **$p < 0.01$, ***$p < 0.001$, one-way ANOVA followed by Tukey's multiple comparison tests, aggregated HTT/vinculin analysis, $F = 19.79$, $p = 0.0023$; mutant exon 1 HTT/vinculin analysis, $F = 49.81$, $p = 0.0002$). **b** HD140Q KI mice were crossed with Cas9 mice to yield KI/Cas9 mice. AAV HTT-gRNA was then injected into the striatum of KI/Cas9 mice at 2 months of age. Eight weeks after injection, the striatum of the injected mice was isolated for analyzing truncated HTT. **c** A fluorescent image of the injected mouse striatum verifying the viral transduction of AAV HTT-gRNA that also expresses red fluorescent protein (RFP). Scale bar: 20 μm. **d** T7E1 analysis of the *HTT* DNAs from the injected region confirming that the only HTT-gRNA, but not control gRNA, could target the *HTT* gene to generate fragmented DNA products. **e** Western blotting analysis of striatum lysates of AAV-injected mice showing that truncation of mutant *HTT* at exon 2 and exon 13 by HTT-gRNAs (HTT-91 gRNA and HTT-571 gRNA) could reduce full-length mutant HTT and increase exon 1 HTT but did not alter the amount of aggregated HTT. **f** mEM48 immunostaining of the AAV-injected striatum of HD140Q KI mice showing that nuclear HTT accumulation after AAV HTT-gRNA injection is not different from that with the AAV control gRNA injection. Scale bar: 20 μm. More than three times of experiments were performed independently for **c**–**f**. Source data are provided as a Source Data file.

inhibitory effects on chaperone activity and account for the preferential accumulation of mutant exon 1 HTT in the striatal neurons.

## Discussion

Using CRISPR/Cas9 to target the *HTT* gene in HD140Q KI mice, we were able to truncate *HTT* to generate N-terminal HTT fragments that are expressed at the endogenous level. These models offer convincing evidence that N-terminal HTT fragments are unable to support the early embryonic development but can cause age-dependent neuropathology in adult mice if they carry a large polyQ repeat. The most interesting finding in our study is that exon 1 HTT or its equivalent fragment containing a large polyQ domain is the critical pathological form. Given the extensive investigation of the cleavage sites of HTT in the regions beyond exon 1[25,26,42,43], our findings offer new insight into the pathogenesis of HD and therapeutic strategies.

Our findings support the previous reports that N-terminal 17 amino acid sequences and polyQ as well polyP domains in exon 1 HTT are dispensable for HTT's functions during early embryonic development[44–46]. However, exon 1 HTT is the major pathogenic form of mutant HTT if it carries a large polyQ repeat. First, all KI mice show the stable N-terminal HTT fragment in the same size as mutant HTT in R6/2 and KI-96 mice which express mutant exon 1 HTT or its equivalent product containing the similar polyQ repeats; second, all different types of KI mice display similar behavioral phenotypes and neuropathology, which are less severe than the reported phenotypes in R6/2 mice that express a much higher level of mutant exon 1 HTT; third, mutant exon 1 HTT is able to preferentially accumulate in striatal neurons to form nuclear aggregates and affect gene transcription in the same manner as KI mice expressing full-length mutant HTT.

It would be interesting to understand why mutant exon 1 HTT is stable in all KI mouse brains. Western blotting with antibodies to the epitopes in exon 1 HTT such as 1C2 for the polyQ domain and mEM48 are able to detect a number of N-terminal mutant HTT fragments in the brains of young and old HD KI mice[13–16]. This fact indicates that proteolytic processing of full-length HTT is mediated by cleavage of HTT at multiple sites and is constantly taking place in young and aged neurons. Different HTT fragments may be continuously cleaved to form the exon 1 HTT product. In the HD brains, mutant exon 1 HTT is more stable because a large polyQ repeat may cause its misfolding and aggregation that is resistant to further digestion. Although mutant exon 1 HTT has no conserved nuclear import sequences, it is able to accumulate in the nucleus because expanded polyQ domain can keep small peptides and cytoplasmic proteins to remain in the nucleus[47]. Striatal neurons may have less capacities to prevent

HTT misfolding so that mutant exon 1 HTT can readily accumulate in their nuclei and form aggregates. Chaperones can reduce protein misfolding but their activities decline during aging[48–50] such that the mutant exon 1 HTT is not efficiently removed and begins to accumulate and form aggregates in aged neuronal cells. In support of this idea, overexpressed mutant exon 1 HTT in R6/2 mice may overwhelm chaperone protection so that mutant exon 1 HTT is broadly accumulated in various brain regions. Consistently, we found that co-chaperone inhibitor HspBP1 is more abundant in the striatal neurons and that depletion of HspBP1 could significantly reduce the nuclear accumulation of mutant HTT in the KI mouse striatum. Once mutant HTT is accumulated in striatal neurons, other striatal neuronal specific factors, such as Rhes, can promote selective neuronal vulnerability in the striatum[51].

Our findings also provide important implications for the pathogenesis of HD and perhaps other polyQ diseases. There has been a large body of literature about protein context-dependent neuropathology in HD and polyQ diseases[52–54]. There is no doubt that protein context determines the function, subcellular localization, stability of proteins, and their association with partners. Our findings indicate that the protein context after exon 1 HTT may not critically modulate HTT toxicity, based on the fact that the newly generated HD KI mice show similar nuclear distribution of mutant exon 1 HTT and develop similar pathological phenotypes. Previous studies have found that caspase-6 cleavage at 586 aa generates N-terminal HTT fragments[25]. However, there are conflicting results about the beneficial effects of blocking caspase-6 cleavage in mutant HTT[28,29]. Because full-length HTT is constantly processed by proteolysis cleavage, it is likely that only those proteins that are able to interact with exon 1 HTT may stably associate with mutant HTT and may be involved in HD pathogenesis. Indeed, many studies have found that posttranslational modulations in exon 1 HTT or altering the first 17 amino acids of HTT can significantly regulate HTT toxicity[55–57].

Using HD KI mice expressing truncated HTT at the endogenous level, we provided strong evidence to support the previous finding that mutant exon 1 HTT is present in different types of HD mice[14,20]. Comparison of the pathological and behavioral phenotypes of HD KI mice expressing full-length or truncated mutant HTT further suggests that mutant exon 1 HTT is a key pathogenic form. Although our RT-PCR analysis showed that mutant exon 1 HTT is mainly generated from the canonical HTT mRNA, aberrant exon 1 splicing can have an important role in generation of this toxic form of HTT when normal HTT-RNA splicing is affected in aged neurons. The findings in the current study also have important implication for treating HD. Targeting mutant HTT by lowering its expression has been well accepted as an effective approach to treat HD. This targeting can be

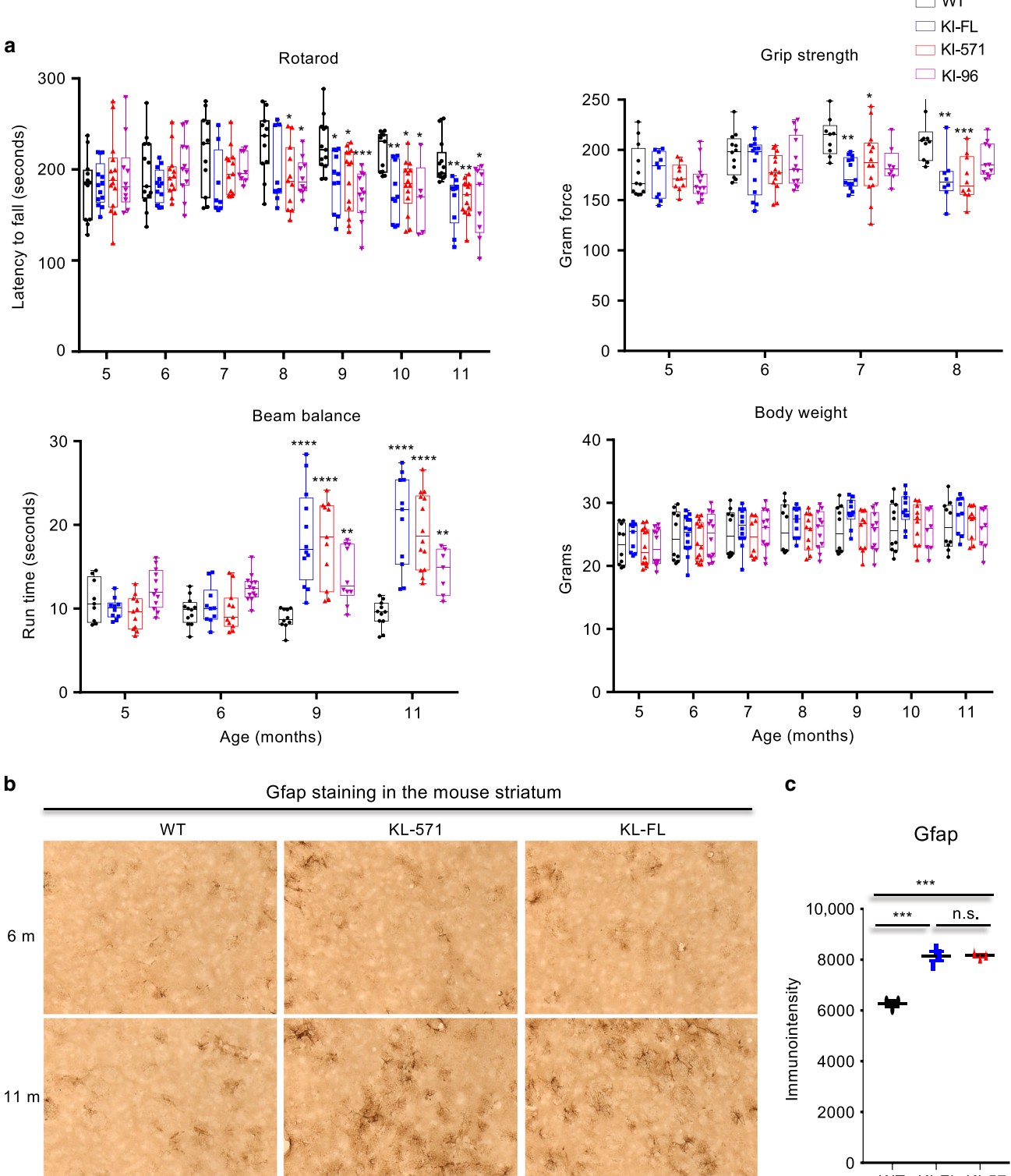

**Fig. 6 Phenotype analysis of different KI mice. a** Comparison of motor function (rotarod performance, balance beam, grip strength) and body weight of wild type and different types of KI (KI-96, KI-571, and KI-FL) mice. Each group consists of 14 mice. Data are presented as minimum to maximum showing all points. *$p < 0.05$, **$p < 0.01$, ***$p < 0.001$, ****$p < 0.0001$, two-way ANOVA followed by Tukey's multiple comparison tests, rotarod analysis, $F = 13.73$, $p < 0.0001$; grip strength analysis, $F = 9.187$, $p < 0.0001$; balance beam analysis, $F = 20.45$, $p < 0.001$; body weight analysis, $F = 4.777$, $p = 0.0029$. Data are presented as mean values ± SEM. **b** Gfap immunostaining of wild type (WT) and KI-571, KI-FL mouse striatum at 6 and 11 months of age. Scale bar: 20 μm. **c** Quantification of relative Gfap immunostaining signals in wild type (WT) and KI-571, KI-FL mouse striatum. The data were obtained from six images in the striatum per mouse ($n = 3$ mice per genotype, ***$p < 0.001$, one-way ANOVA followed by Tukey's multiple comparison tests, $F = 65.67$, $p < 0.0001$). Data are presented as mean values ± SEM. Source data are provided as a Source Data file.

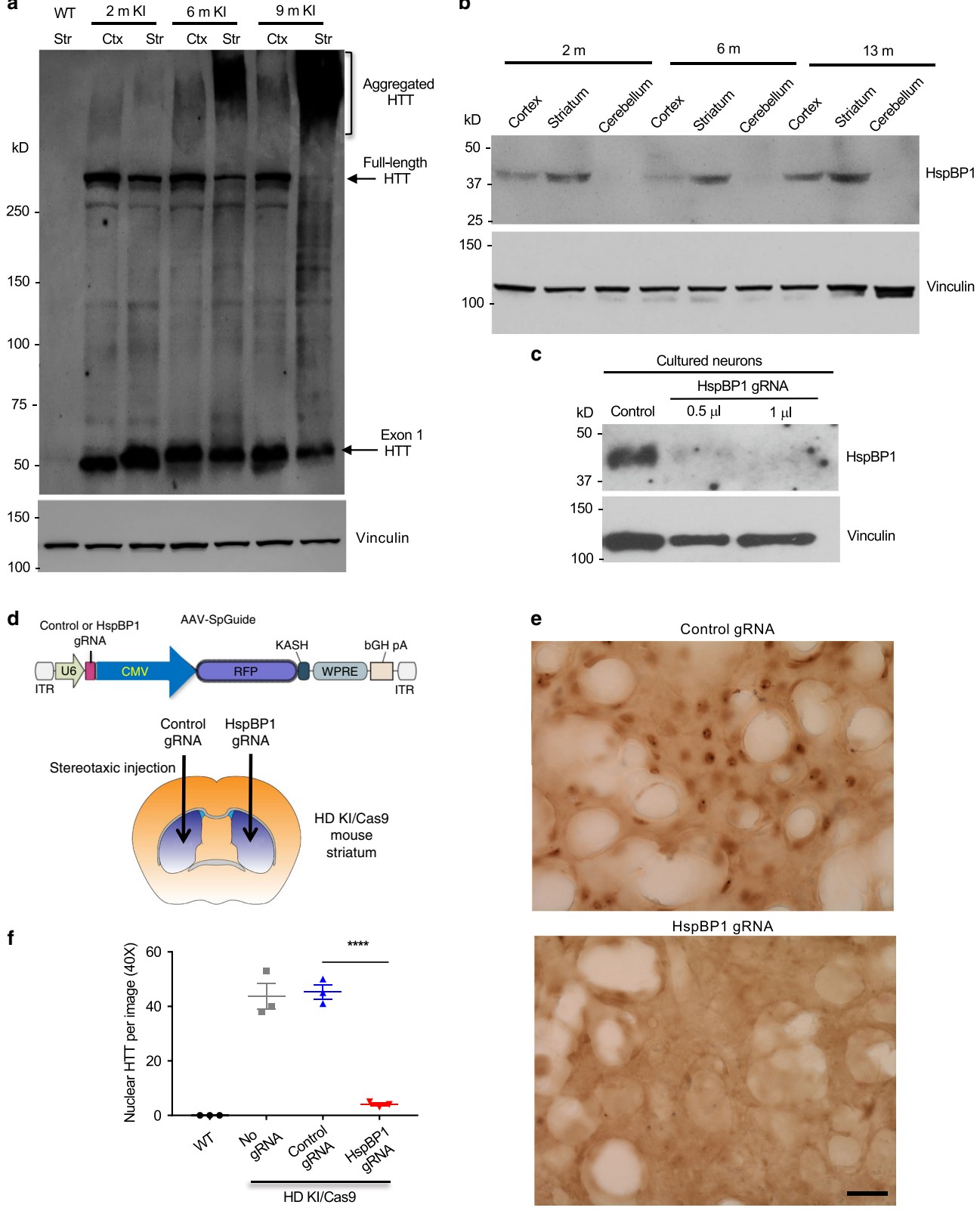

achieved by using antisense oligonucleotides, siRNA, or CRISPR/Cas9[19,33,58]. Because mutant exon 1 HTT can be constantly generated by proteolytic cleavage and aberrant splicing of exon 1[20], genetic targeting should be focused on the production of exon 1 HTT. Alternatively, drugs or chemicals that act on exon 1 HTT are more likely to achieve effective therapeutic effects than those acting on other regions of HTT.

## Methods

**Antibodies and reagents**. See Supplementary Tables 1–4.

**Mice maintenance and breeding**. Mice (C57BL/6) were bred and maintained in the animal facility at Emory University under specific pathogen-free conditions in accordance with institutional guidelines of the Animal Care and Use Committee at Emory University. All mice were maintained on a 12:12 h light/dark cycle (lights off at 7 p.m.). The temperature was maintained at 22 ± 1 °C with relative humidity

**Fig. 7 HspBP1 expression accounts for the nuclear accumulation of mutant HTT in KI mouse striatum. a** mEM48 western blot analysis of the cortex and striatum of HD140Q KI mice at different ages revealing that mutant exon 1 HTT is constantly expressed but selectively forms aggregates in the aged striatum. More than three times of experiments were performed independently. **b** Western blotting of wild-type mouse brains showing that HspBP1 is more abundant in the striatum and its expression increases with age. **c** Western blotting of cultured mouse striatal neurons showing that targeting HspBP1 by AAV Cas9/HspBP1-gRNA could effectively reduce HspBP1 expression. **d** Injection of AAV control gRNA and AAV HspBP1-gRNA into the striatum of KI/Cas9 mice at 25 weeks of age. The mouse brain image was remixed from the clipart https://commons.wikimedia.org/wiki/File:Mouse_Dorsal_Striatum.pdf, under the Creative Commons Attribution-Share Alike International license. **e** Eight weeks after injection, mEM48 immunostaining of the injected striatum showing that deletion of HspBP1 could markedly reduce the nuclear accumulation of mutant HTT. Scale bar: 40 μm. **f** Quantitative analysis of nuclear HTT staining in the AAV-injected mouse striatum. The data were obtained from five images in the striatum per mice ($n = 3$ mice per group, ****$P < 0.0001$, one-way ANOVA followed by Tukey's multiple comparison tests, $F = 82.84$, $p < 0.0001$). Data are presented as mean values ± SEM. Source data are provided as a Source Data file.

(30–70%). The studies followed the protocol approved by the Animal Care and Use Committee at Emory University.

**Generation of truncated HD KI mice**. Generation of mice expressing truncated normal HTT (d177 and KI-1367) was performed by Beijing Biocytogen, China. Briefly, gRNAs were designed to target different exons in the mouse *HTT* gene (Supplementary Table 3). Cas9 mRNAs and gRNAs were mixed for pronuclear injection into the mouse zygotes (C57BL/6). The mice of generation F0 were genotyped with the PCR primers specific for the targeting exons (Supplementary Table 4). Mice carrying the disruption of *HTT* exons were used to breed to generate F1 and F2 generation mice at Beijing Biocytogen, China.

Generation of N-terminal mutant *HTT* knock-in mice (KI-96 and KI-571) was performed by Gene Edit Biolab, Morehouse School of Medicine, Atlanta, USA. The mouse zygotes of HD140Q KI mice were injected with mixed mRNA of Cas9 and gRNA (1 μl mixture per zygote). Founders were imported to Emory animal facility and kept in a quarantine room before releasing to a regular housing room. CRISPR/Cas9-targed male mice were crossed with wild-type female mice to generate F1 mice. Mice that carried the desired mutations were crossed with wild-type mice to generate heterozygous KI mice. Full-length mutant HD140Q KI mice were generated as described previously[9] and maintained in the animal facility at Emory University for our previous studies[33]. All animal experiments were approved by Division of Animal Resources of Emory University in accordance with institutional guidelines of the Animal Care and Use Committee at Emory University.

**Generation of Cas9 and KI/Cas9 mice**. Cre-dependent cas9 transgenic mice[31] were obtained from the Jackson Laboratory (Stock No: 024857, Rosa26-LSL-Cas9 knock-in). To activate Cas9 expression, the Cre-dependent Cas9 mice were crossed to EIIA-Cre transgenic mice to generate Cas9 mouse line that ubiquitously express Cre in all tissues. The constitutive Cas9-expressing mice were fertile, and the homozygosity was viable. Furthermore, the Cas9 mice had normal litter sizes and similar body weight compared to wild-type mice and showed no morphological abnormalities, in consistence with previous studies[31]. Germline transmissible Cas9 mice were isolated after more than 3 generations, which were then crossed with HD140Q KI mice to generate KI/Cas9 mice that express both mutant HTT and Cas9. KI/Cas9 mice allowed for creating mutations in the mutant *HTT* allele by expressing only HTT-gRNA via AAV stereotaxic injection. Primers used for genotyping Cas9 transgenic mice and KI/Cas9 mice were listed in Supplementary Table 4. The expression of mutant HTT and Cas9 protein were confirmed by western blot analysis.

**Genotyping, T7E1 assay, and sequencing**. Genomic DNA was isolated from mouse tails, and PCR genotyping with primers that flank the CAG region was used to identify HD mice that carry expanded polyQ repeats. The sequences of primers mentioned above are indicated in Supplementary Table 4. The *HTT* gene domain containing the targeted sites by CRISPR/Cas9 was amplified by PCR and subjected to T7E1 assay for detection of mismatch mutation between edited gene alleles and wild-type gene alleles. PCR products were obtained using the condition consisting of denaturing at 95 °C for 2 min, decelerating 2 °C/s to 85 °C, decelerating 0.1 °C/s to 25 °C, and holding at 16 °C for 10 min. For T7E1 assay, 5 μl of the annealing PCR product was treated with 0.2 μl T7 Endonuclease I (NEB, #M0302S) in a 20 μl system and run on 1.5–2.0% agarose gel after incubation at 37 °C for 30 min. Founders (F0) with positive T7E1 assay results were bred to F1 generation mice, which were regularly tested by genotyping and T7E1 assay. PCR products that showed mutation specifically at the mutant allele with expanded CAG/polyQ repeats were isolated by Topo-TA cloning (ThermoFisher, #450425) for sequencing to identify the insertion or deletion (indel) mutation induced by CRISPR editing. F1 mice with positive indel mutation at expanded polyQ allele were bred to F2 generation. Germline transmissible indel mutation in F2 generation was confirmed by T7E1 assay and Topo-TA cloning described as above. After crossing mice multiple times, we obtained one positive HD KI-571 founder, three positive HD KI-96 founders with germline stable indel mutation at expanded CAG/polyQ

alleles. The gene editing efficiency and transgene expression were also determined via western blotting analysis.

**Total RNA extraction, RT-PCR, and qPCR**. Frozen or fresh mouse brains were added with trizol (0.5 ml for 20 mg brain tissue) before homogenization. The homogenate was added with 200 μl chloroform and thoroughly mixed by vortex and then incubated at room temperature for 5 min. After centrifugation at $10,000 \times g$ for 15 min at 4 °C, the upper aqueous phase was transferred to a new Eppendorf tube. After adding 0.5 ml isopropanol and incubated at room temperature for 10 min, the mixture was centrifuged at $10,000 \times g$ for 10 min. DNase treatment was performed each time and the DNase-treated RNA was purified by ethanol precipitation. The total RNA was quantified by nanodrop and 2 mg of total RNA was used for reverse transcription reactions using the High-Capacity RNA-to-cDNA™ Kit (Applied Biosystems, 4368814) and oligo dT primer (Invitrogen, 18418020). The control was the same RNA samples without reverse transcriptase. The synthesized cDNA (1 μl) was combined with 10 μl Power SYBR™ Green PCR Master Mix (Applied Biosystems, 4367659) and 1 μl of each primer in a 20 μl reaction when performing qPCR. The reaction was performed in the Eppendorf, Realplex Mastercycler thermocycler using primers shown as Supplementary Table S4. Acquired images from RT-PCR were subjected to densitometric quantitation using ImageJ software. The relative values of PCR results were normalized by GAPDH levels prior to calculations by graphpad prism7. For qPCR analysis, $2^{\Delta\Delta Ct}$ method was used to calculate the relative fold gene expression of samples. In detail, $\Delta Ct_{sample} = Ct_{gene\ of\ interest} - Ct_{housekeeping\ gene}$, $\Delta\Delta Ct = \Delta Ct_{sample} - \Delta Ct_{calibrator}$ (In Supplementary Fig. S8g and S8h, the average value of $Ct_{housekeeping\ gene}$ of exon 1–intron 1 in cortex of KI mice was seen as $\Delta Ct_{calibrator}$). Relative expression level = $2^{-\Delta\Delta Ct}$, which was used for further analysis by graphpad prism7.

**Western blotting analysis, immunohistochemistry, and immunofluorescence staining**. For western blotting analysis, mouse brain tissues were lysed in ice-cold RIPA buffer (50 mM Tris, pH 8.0, 150 mM NaCl, 1 mM EDTA pH 8.0, 1 mM EGTA pH 8.0, 0.1% SDS, 0.5% DOC, and 1% Triton X-100) containing Halt protease inhibitor cocktail (ThermoScientific) and PMSF. The lysates were incubated on ice for 30 min, sonicated, and then equal amounts of proteins from the whole lysates determined by BCA assay were resolved by 4–12% or 4–20% polyacrylamide Tris-Glycine gels (Invitrogen) and subjected to western blotting analysis with appropriate primary antibodies (1:50 dilution for mEM48). To quantify the ratios of interesting bands to the loading control, the western blots were cut to strips containing the interesting bands or the loading control protein for probing with their antibodies so that the ratios could be obtained from the same western blots.

For immunohistochemistry, mice were anaesthetized with peritoneal injection of 10% euthasol (39.0 mg/ml pentobarbital sodium and 5 mg/ml phenytoin sodium) and perfused with 0.9% NaCl followed by 4% paraformaldehyde (PFA). The brains were removed and post-fixed in 4% PFA overnight at 4 °C. The brains were transferred to 30% sucrose for 48 h to let the brain completely sink to the bottom of the tube, and then cut to 20 or 40 μm sections with the cryostat (Leica CM1850) at −20 °C. Mouse tissue sections were permeabilized with 0.3% Triton X-100/PBS at room temperature for 1 h and then treated for antigen retrieval with citric acid buffer (10 mM citric acid, 0.05% Tween 20, pH 6.0) at 95 °C water bath for 10 min. The mouse tissue sections were then blocked with 0.1% Triton X-100/2%, normal goat serum (NGS)/3%, and BSA/1× PBS for 30 min. For immunohistochemistry with DAB staining, 40 μm sections were incubated with anti-HTT antibody mEM48 (1: 100 dilution) for at least 48 h at 4 °C. Avidin-Biotin Complex kit (Vector Laboratories) and DAB kit (Sigma, D4418) for immunodetection were used. Images were acquired with a Zeiss microscope (Carl Zeiss Imaging, Axiovert 200 MOT) with a digital camera (Carl Zeiss Imaging, AxioCam HRc) and AxioVision software. For immunofluorescent staining, 20 μm sections were incubated with primary antibodies in the same buffer at 4 °C overnight. After washing with 1× PBS, the sections were incubated in fluorescent secondary antibodies and Hoechst stain. Fluorescent images were acquired with a Zeiss microscope (Carl Zeiss Imaging, Axiovert 200 MOT) and a digital camera (Carl Zeiss Imaging, AxioCam HRc) using AxioVision software.

**AAV viral preparation and stereotaxic injection**. CRISPR/Cas9-expressing viral vector (PX552) was obtained from Addgene (plasmids #60958). AAV HTT-gRNAs were generated by inserting gRNAs into PX552 via *Sap*I restriction sites. These viral vectors were sent to the Viral Vector Core at Emory University for packaging and purification of AAV9 or AAV2/9 viruses. The genomic titer of viruses was determined by PCR method. AAV HspBP1 vector and viruses were generated previously in our study[41].

Stereotaxic injection of AAV into mouse brains was performed using the method described in our previous studies[33]. Briefly, mouse at 2–3 months of age was anesthetized using isoflurane (2.5% sedation, 1.5% afterwards) delivered with a vaporizer and stabilized in a stereotaxic instrument (David Kopf Instruments). All surgical procedures were performed in a designated procedure room and in accordance with the Guidelines for the Care and Use of Laboratory Animals and biosafety procedures at Emory University. Viruses expressing gRNA were injected into one side of the mouse striatum according to the following coordinates: 0.55 mm rostral to bregma, 2.0 mm lateral to the midline, at 3.5 mm ventral from the dural surface. The injection was conducted using a Hamilton syringe and a syringe infusion pump (World Precision Instruments, Inc., Sarasota, Florida, USA) at a speed of 200 nl/min over a period of 5 min. Mouse was administered Meloxicam (5 mg/kg/day SC injection) pre-operation for pain management. During the surgery, mouse was monitored with breath rates, movement, and body temperature continuously. After the surgery, mouse was transferred into a filter top covered clean cage, which was situated on top of a controlled heating pad (35 °C) and was monitored every 15 min until the mouse was completely recovered from the anesthesia and could move freely. After injection 2–3 months, the mouse was killed by deep anesthesia with inhalation of isoflurane or peritoneal injection of 10% euthasol (39.0 mg/ml pentobarbital sodium and 5 mg/ml phenytoin sodium), and their brain tissues were isolated for western blotting and immunohistochemical or immunofluorescent analysis.

**Mouse behavioral studies**. All animal tests were performed in accordance with NIH guidelines for procedures and approved by the Institutional Animal Care and Use Committee of Emory University. At least 8 mice per genotype and both male and female mice were included for behavioral studies. Mouse behavior was assessed using a rotarod apparatus (Rotamex 4/8, Columbus Instruments International). Mice were trained on the rotarod at 5 RPM for 5 min for three consecutive days. For testing, the rotarod gradually accelerated to 40 RPM over a 5-min period. Latency to fall was recorded for each trial. Each mouse went through three trials, and the average data of each group charted ($n > 8$ mice per group). The balance beam apparatus consists of 1 m beams with a flat surface of 6 mm width resting 50 cm above the table top on two poles. A black box at the end of the beam is the finish point. The time required for a mouse to cross to the center (80 cm) is measured as previously described[33]. For the gripping ability test, mice were measured by grip strength meters (Ametek Chatillon) every month as previously described[33]. The peak grip strength observed in five trials was recorded. Mouse body weight was measured monthly.

**RNAseq analysis**. The striatum of WT, KI-96, KI-571, and KI-FL mice at 6 months of age were used to extract RNAs, which were then subjected to RNAseq analysis by Novogene. Each genotype consisted of 4–5 mice. Sequencing adapters were trimmed by BBduk (minlen = 100 ktrim = r $k$ = 23 mink = 11 hdist = 1 tpe tbo; http://jgi.doe.gov/data-and-tools/bb-tools/). Reads with length under 100 bp after trimming were discarded. Clipped reads were aligned to mouse genome GRCm38 (primary assembly) by STAR aligner with default settings. Read counts for genes were obtained by using HTSeq. All mRNA differential expression analyses were performed in R using the Bioconductor package DESeq2 version 1.24.0. We removed one WT sample identified as a potential outlier and used the default settings with 'sac batch' as a covariate variable. We tested differential expression in two-sample tests of KI (KI-FL, KI-96 and KI-571) samples vs. wild-type (WT) samples.

To draw heatmap, we first applied the variance-stabilizing transformation implemented in DESeq2 to raw counts and removed the effect of 'sac batch' by using function 'removeBatchEffectRaw' in Bioconductor package Limma. Then, sample means corresponding to the four genotypes were calculated based on these transformed values. Finally, heatmap of normalized sample mean values of the four genotypes were generated for genes in selected modules (M2, M7, M9, M20, M25, M39, M43, and M46)[37]. For differential expression (DE) tests of different KI genotypes vs. WT (KI-FL vs. WT, KI-96 vs. WT and KI-571 vs. WT), in each selected module M2, M7, M9, M20, M25, M39, M43, and M46, genes were divided into two groups by checking whether corresponding log fold change greater than 0 or not. The bar plot was generated based on averages of log10 $p$-value of these grouped genes in different modules for different DE tests (KI-FL vs. WT, KI-96 vs. WT and KI-571 vs. WT).

**Statistics and reproducibility**. For biochemical and histological studies, we used at least three mice per group. For animal behavioral studies, we used at least eight mice per genotype. More than three independent experiments were done to obtain the gel, blot, or micrograph results that were used for figure presentations, and the representative results were shown in figures. Statistical significance was assessed using the two-tailed Student's *t*-test for comparing if there were only two groups. When analyzing multiple groups, we used one-way ANOVA or two-way ANOVA followed by Tukey's multiple comparisons test to determine statistical significance. A *p*-value < 0.05 was considered significant. Data represent the mean ± SEM. Calculations were performed with GraphPad Prism7 software.

**Reporting summary**. Further information on research design is available in the Nature Research Reporting Summary linked to this article.

## Data availability

The data generated or analyzed during the current study are available within the article, supplementary information, attached source data file, and from the corresponding author upon reasonable request. The RNA sequencing data have been deposited in the Gene Expression Omnibus (GEO) database and the accession code is GSE145879.

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

## Acknowledgements

This work was supported by the NIH of the United States of America (NS036232, NS101701, NS095279, and NS095181), The National Key Research and Development Program of China Stem Cell and Translational Research (2017YFA0105102), National Natural Science Foundation of China (81830032; 31872779), and Key Field Research and Development Program of Guangdong province (2018B0300337001). We thank Renbao Chang and Yinghui Zheng for technical assistance in the initial studies of d177 KI mice, and the Emory University Viral Core Facility for generating AAV viruses.

## Author contributions

The author contributions in this manuscript were as follows: H.Y., S.L., and X.-J.L. designed the research experiments. H.Y., S.Y., L.J., L.H., W.Y., Y.P., X.Z., and P.Y. performed the experiments, L.C. and Z.Q. analyzed RNAseq data. H.Y. and X.-J.L. wrote the manuscript. S.L. and S.Y. edited the manuscript.

## Competing interests

The authors declare no competing interests.
