## [Peer Review File · Nature Communications]

Reviewers' comments:

Reviewer #1 (Remarks to the Author):

Using a CRISPR/Cas9-based editing strategy, Yang et al., have generated a series of mutant Huntingtin N-terminal knock-in truncation mutations to both characterize mutant Huntingtin N-terminal fragments in HD mouse model pathogenesis, and to understand the mechanism of selective neuropathology in HD. Based on their findings reported in this manuscript, together with past work from the laboratory implicating neuronal expression of HspBP1 in the preferential accumulation of mutant Huntingtin in neuronal cells but not in glia, the authors now also provide evidence suggesting that the abundant striatal expression of HspBP1 in the HD model mice is responsible for the preferential accumulation of mutant Huntingtin in the striatum.

Although there is abundant prior evidence in the field for the importance of N-terminal fragments of mutant Huntingtin in contributing to the mechanism of HD pathogenesis, the novel knock-in truncation mutants generated by Yang et al. express the different mutant Huntingtin fragments under endogenous promoter control with normal developmental and tissue-specific expression levels. Strong data is providing demonstrating that HD model pathogenesis in all the truncation mutants is almost indistinguishable from the parent full-length mutant Huntingtin knock-in model, and that regardless of where the truncation is made, all models produce equivalent levels of a mutant Htt N-terminal species corresponding to an exon-1 fragment. The authors also provide evidence suggesting that high levels of the HspBP1 co-chaperone in the striatum are responsible for the accumulation of the mutant Huntingtin exon-1 fragment in this brain region, and that AAV-mediated Cas9/HspBP1 gRNA targeting of HspBP1 expression in the striatum reduces the nuclear accumulation of the mutant Huntingtin fragment. Statistical analyses are appropriate, and sufficient information is provided to allow reproduction of the work in other laboratories.

The findings of Yang et al. will be of interest to both the HD community and to a wider audience, as they provide new data supporting the importance of the mutant Huntingtin exon-1 fragment in HD pathogenesis, and a new hypothesis explaining how tissue- and cell-specific expression of a critical co-chaperone can contribute to brain region and cell-specific neurodegeneration. Importantly, Yang et al. provide additional evidence for the field that targeting the generation and/or stability of the mutant Huntingtin exon-1 fragment may be critical for development of the next generation of HD therapeutics.

Two of the authors main conclusions, however, could be strengthened with some additional data. To evaluate how much aberrant exon1-intron1 levels contribute to the total levels of the mutant Huntingtin exon-1 fragment, the authors perform an RT-PCR experiment comparing the levels of normally spliced exon1-exon3 RNA and aberrant exon1-intron1 RNA. They find that the exon1-exon3 PCR product was much more abundant than the exon1-intron1 product. Based on this result, the authors conclude that the mutant Huntingtin exon-1 fragment is likely produced by proteolytic cleavage of full-length mutant Huntingtin or longer mutant Huntingtin fragments. This reasoning is potentially incorrect, as the western data using Huntingtin antibodies other than the mEM48 antibody (which may also be recognizing the neoepitope generated by formation of the exon-1 truncation event and therefore not reflect fragment abundance) also show very low levels of the ~50 kD species corresponding to the exon-1 fragment (Supplementary Fig. 5). In addition to the mEM48 antibody, the authors could consider using the S830 antibody that also preferentially recognizes the mutant Huntingtin exon-1 fragment (Sathasivam et al., 2013). The MW8 antibody, like mEM48 recognizes mutant Huntingtin aggregates but it is difficult to tell in Supplementary Fig. 5 western blot if a ~50 kD species is present because the exposure appears to be less than the mEM48 blot in this figure. Formic acid treatment of the SDS-insoluble material in the samples may also enhance the detection of the mutant Huntingtin exon-1 fragment using the anti-N17, anti-polyQ antibodies, and MW8 antibodies. Comparison of the authors RNASeq data to human Huntingtin exon-1 sequence may also help to identify levels of exon1-intron1 RNA transcripts in their data sets.

The authors' conclusion that region-specific expression of HspBP1 is responsible for the accumulation of mutant Huntingtin could be strengthened by performing immunohistochemistry to complement the western data that is provided in the manuscript. In the HD knock-in models, mutant Huntingtin nuclear accumulation and aggregation also occurs in cortical layer V, in the hippocampus (of older mice), and in the piriform cortex. Detection of enhanced immunohistochemical staining of HspBP1 in these brain regions can provide complementary data strengthening the authors' hypothesis. Alternatively, data mining of single-cell RNASeq or region-specific RNASeq databases may also provide additional confirmation provided that the cell and region-specific elevation of HspBP1 is under transcriptional control.

Minor comments and questions:

1. Line 295: "...Htt is prone to misfold and that chaperone is important.." should be "...Htt is prone to misfold and that chaperones are important..." Line 297: "effects on chaperone and account" should be "effects on chaperone activity and account"
2. Mouse behavioral studies: were both males and females used in the experiments?
3. Legend to Fig. 5b: Please include the antibody used for this blot (mEM48?).
4. Legend to Fig. 7a: please include the antibody used for this blot. Fig. 7b: Are Hsp70 and CHIP levels also affected by aging?

Sincerely,
Scott Zeitlin

Reviewer #2 (Remarks to the Author):

The document entitled 'Truncation of mutant huntingtin in knock-in mice via CRISPR-Cas9 uncovers exon1 huntingtin as a key pathogenic form' by Yang et al. makes use of CRISPR-Cas9 to produce knock-in mouse models expressing various truncated versions of the mutant huntingtin protein. The nuclear aggregation and expression of exon1 is compared between the knock-in lines of mice. These studies indicate the truncation of mHtt does not alter the production or localization of exon1. Behaviour experiments and assessment of neuropathology further indicate that huntingtin of various lengths has the same degree of pathogenicity. Finally the authors show that nuclear aggregation can be prevented by inhibiting HspBP1.

General comments

- Rationale for the study is interesting but the characterization of the behaviour and neuropathology of the mice is very superficial which prevents firm conclusions about the impacts of truncating mHtt
- The figures could use further polishing and contain many crocked lines and titles that are not centred
- Many figures are lacking quantifications that would be essential to support the claims stated in the text
- The story is difficult to follow and some of the figure panels are redundant
- The N for biochemistry experiments is very low (N=3)

Major concern

1. In Figure 1b for Exon-D the numbers of animals in the three groups does not add up to the total listed (6+7+12 =25 not 39).

2. Figures 1 and 2 indicate that one of the studied proteins was produced by targeting Exon 13 (KI-571) but in figure 5 it is stated that KI-571 is produced by cleavage of exon 6.
3. Explanations of the rationale for cleaving at Exons 2 and 6 are missing. Given that one of the purposes of this study is to understand if the various cleavage products of mHtt contribute to disease pathology it would be useful to have further details about how the various Knock-in lines impact the cleavage of mHtt.
4. In Figure 2b and d western blots of KI-571 and KI-96 are shown which indicate that these mice do not have full length mHtt but they do express exon1 mHtt. I find it odd that a longer molecular weight band is not detected. The authors should discuss this.
5. Figures 2e, 3b-c and 4a all show aggregation in the striatum and are redundant
6. In the results section it is stated the KI-96 has a worse score on the balance beam than the other knock-in lines but this is not indicated with statistics on the graph and is inconsistent with what is shown at later time points where this model seems to have improved performance. This data should be discussed further.
7. To conclude that the truncation of mHtt does not impact neuropathology and behaviour a much broader range of assessments would be required. Astrogliosis is far from the most apparent or important neuropathological hallmark of disease and cognitive and non-motor performance are also important aspects of pathology.
8. While the data on HspBP1 is interesting it seems rather disjointed from the main focus of the paper.

Minor concerns

1. For Figure 6 a two-way ANOVA was performed to assess all behaviour data. However a repeated measures ANOVA would be more appropriate if the same animals were used at multiple time points.
2. In the results section titled Preferential nuclear accumulation of exon1 like Htt in the striatum many of the references to figures seem to be inaccurate (Lines 169, 170, 182 and 184)
3. In Figure 2e data is not shown for KI-96 at 11 m. Given that these mice appear to have a different behavioural phenotype it would be important to see the nuclear aggregation at later ages.

Verdict: Reject

Response letter

Below are point-by-point responses (in blue) to each issue of the reviewers. Important changes are highlighted in yellow in the manuscript text.

Reviewer #1 (Remarks to the Author):

The findings of Yang et al. will be of interest to both the HD community and to a wider audience, as they provide new data supporting the importance of the mutant Huntingtin exon-1 fragment in HD pathogenesis, and a new hypothesis explaining how tissue- and cell-specific expression of a critical co-chaperone can contribute to brain region and cell-specific neurodegeneration. Importantly, Yang et al. provide additional evidence for the field that targeting the generation and/or stability of the mutant Huntingtin exon-1 fragment may be critical for development of the next generation of HD therapeutics.

Two of the authors main conclusions, however, could be strengthened with some additional data. To evaluate how much aberrant exon 1-intron1 levels contribute to the total levels of the mutant Huntingtin exon-1 fragment, the authors perform an RT-PCR experiment comparing the levels of normally spliced exon 1-exon 3 RNA and aberrant exon 1-intron1 RNA. They find that the exon 1-exon 3 PCR product was much more abundant than the exon 1-intron1 product. Based on this result, the authors conclude that the mutant Huntingtin exon-1 fragment is likely produced by proteolytic cleavage of full-length mutant Huntingtin or longer mutant Huntingtin fragments.

1. This reasoning is potentially incorrect, as the western data using Huntingtin antibodies other than the mEM48 antibody (which may also be recognizing the neoepitope generated by formation of the exon-1 truncation event and therefore not reflect fragment abundance) also show very low levels of the ~50 kD species corresponding to the exon-1 fragment (Supplementary Fig. 5). In addition to the mEM48 antibody, the authors could consider using the S830 antibody that also preferentially recognizes the mutant Huntingtin exon-1 fragment (Sathasivam et al., 2013).

The reviewer raised an important issue of whether mEM48-reacted band (the ~50 kD) is abundant because other antibodies failed to detect it. We believe that the specific conformation of this N-terminal HTT fragment allows mEM48 to recognize it on western blots, as the epitope of mEM48 is VA residues after the polyP domain, which is different from the epitopes for other antibodies, and expanded repeats can make this epitope recognizable by mEM48 (Wang et al., 2008). In our recent publication (Fig. 1D in Yang et al., PNAS 2020), we also showed that this 50 kD band is equivalent to exon 1 mutant HTT in full-length HD KI mice.

Fig. 1D in Yang S et al., PNAS 2020.

As indicated by the reviewer, our RNA analysis has already shown that the transcript of exon 1 HTT is expressed at the same level as the endogenous HTT transcript. To examine the expression of the exon 1 HTT at the protein level, we also performed immunocytochemical staining with two different antibodies, 1C2 and mEM48. We now provided new results to support the abundance of exon 1 HTT in the striatum of KI-571 mouse brain. Under the same immunohistochemical staining conditions, both 1C2 and mEM48 immunostaining revealed comparable levels of nuclear HTT and aggregates in KI-571 and KI-FL mouse striatum (Supplemental Fig. S6). However, the ~50 kD band on Western blots was only recognized by mEM48, but not 1C2, again suggesting that this N-terminal HTT band on the blot has a specific conformation that is recognized only by mEM48. Indeed, many anti-HTT antibodies have been generated, but only a few of them can recognize either soluble N-terminal HTT or aggregated HTT on Western blots. This fact fits with the idea that polyQ expansion can alter the conformation of N-terminal HTT fragments.

As for the use of S830, this is an antibody for exon 1 HTT but is not commercially available. In addition, in Landles et al's study that compared several antibodies to HTT via western blotting of HD KI mouse brain (Fig. 1 in Landles et al., JBC 2010), the immunoblotting pattern of S830 is similar to that of 1C2 and did not show the obvious ~50 kD band. Only after immunoprecipitation of mutant HTT, S830 was able to recognize exon 1 HTT in different types of HD mouse models (Sathasivam et al., 2013). We have tried several antibodies that were raised against exon 1 HTT but could not detect the ~50 kD band on Western blots without immunoprecipitation (Supplementary Fig. 5). All these indicate that N-terminal HTT with a large polyQ repeat has a unique conformation that can be readily detected on the blots by a specific antibody like EM48. We have included this explanation in the revision as follows:

“However, compared with HD140Q knock-in (KI-FL) mouse brain in which full-length mutant HTT was expressed and a series of N-terminal fragments were visible, only one HTT fragment around 50 kD was seen in the brain of KI-571 mice. It is likely that this N-terminal HTT fragment is stable in KI mice and possesses a unique conformation that can be readily recognized by mEM48. In support of this idea, immunohistochemical staining of the striatum of KI-571 and KI-FL mice with different antibodies (1C2 and mEM48) showed the comparable levels of 1C2 and

mEM48 immunoreactive nuclear HTT and aggregates (Supplementary Fig. 6). It has been well documented that small N-terminal HTT fragments are able to accumulate in the nucleus to form aggregates^{22,23}. Thus, the ~50 kD band seen on the western blots is likely present in both KI-571 and KI-FL mouse striatum to accumulate in the nucleus but its unique conformation only allows mEM48 to detect it via Western blotting. The size of this HTT fragment is similar to exon 1 HTT that was found in the brains of different HD mice after immunoprecipitation of mutant HTT^{14,20}.”

2. The MW8 antibody, like mEM48 recognizes mutant Huntingtin aggregates but it is difficult to tell in Supplementary Fig. 5 western blot if a ~50 kD species is present because the exposure appears to be less than the mEM48 blot in this figure. Formic acid treatment of the SDS-insoluble material in the samples may also enhance the detection of the mutant Huntingtin exon-1 fragment using the anti-N17, anti-polyQ antibodies, and MW8 antibodies.

The epitope of MW8 is different from that for mEM48 (see Supplementary Fig. 5). Using a longer exposed blot probed with MW8, we still could not detect the ~50 kD band that should be specifically present in KI-571 mouse striatum when comparing with WT mouse striatum (Supplementary Fig. 5).

As for formic acid treatment, our previous studies have already shown that formic acid would dissolve EM48-reactive products while 1C2 would only reveal a smear of mutant HTT (Zhou et al., J Cell Biol. 2003 Fig 2, see below). Our preliminary experiments with formic acid treatment did not yield any positive results so that we did not pursue this experiment, because formic acid treatment could alter protein conformation and the quality of protein samples. Since mEM48 immunoblotting allowed us to reliably detect exon 1 HTT on Western blots without any treatment, we used mEM48 for further characterization of exon 1 HTT in the newly established KI mouse brains.

Figure 2. (B) The nuclear fractions of the striatum from control (C), HD, and Alzheimer's disease (AD) were dissolved with formic acid (FA) and resolved by SDS-PAGE. Products immunoreactive to 1C2 were not seen without FA treatment (-FA) but appeared as a smear in the HD sample after FA treatment (+FA). (C) FA treatment reduced the amount of EM48 or ubiquitin-labeled aggregates (arrow).

3. Comparison of the authors RNASeq data to human Huntingtin exon-1 sequence may also help to identify levels of exon 1-intron1 RNA transcripts in their data sets.

This is a good suggestion. We also followed the reviewer's suggestion to use RNAseq to analyze the expression of the incomplete spliced form. In KI-FL and KI-96 mice, we counted the number of reads from regular transcription and aberrant splicing, and calculated the ratio. Our results indicate that the aberrantly spliced mutant *HTT* mRNAs are indeed in the KI mouse striatum. We also found that mutant *HTT* mRNAs derived from normal splicing were more abundant than those derived from aberrant splicing.

However, if the large CAG repeats in the *HTT* gene make the exon 1-intron 1 DNA more stable during RNA extraction, this assay may not be able to rule out the influence from genomic DNA amplification when such DNA has not been completely removed.

To ensure that no genomic DNAs are amplified by RT-PCR, we used two rigorous controls: one was to perform RT-PCR to amplify both exon 1-exon 3 and the incomplete spliced exon 1-intron 1 in the same PCR reaction with or without reverse transcriptase, the other was to treat mRNA samples with DNAase to remove genomic DNAs before performing real time PCR and then compare the relative levels of exon 1-exon 3 and exon 1-intron 1 under the same PCR conditions. The results show that the canonical splicing product (exon 1-exon 3) is more abundant than the incomplete splicing exon 1-intron 1, supporting our conclusion that mutant *HTT* exon 1 fragment is mainly caused by proteolytic cleavage. However, we also verified that the incomplete spliced *HTT* mRNA was only seen in the KI mouse striatum. Based on these findings, we provided new Supplementary Fig. 8 and reworded the description and interpretation of the results as follows:

"Exon 1 *HTT* fragment was found to be generated by aberrant exon 1-intron 1 RNA, which was observed only in the HD brains in which a large CAG repeat is present^{20,24}. Using RT-PCR of WT and KI mouse, we confirmed the selective expression of aberrant exon 1-intron 1 RNA in the KI mouse striatum when compared with WT mouse striatum (Supplementary Fig. 8a, b). By examining RNAseq results, we also found that aberrant exon 1-intron 1 RNA was present in the KI mouse striatum, but its level appeared to be lower than the normally spliced mutant *HTT* mRNA (Supplementary Fig. 8c). To more rigorously compare the levels of canonical *HTT* mRNA (exon 1- exon 3) and aberrantly spliced mRNA (exon 1-intron 1), we performed RT-PCR to detect their levels in the same PCR reactions by including a critical control without reverse transcriptase to ensure that PCR products seen in the mouse brains are specifically derived from cDNAs, rather than genomic DNAs. Using this assay, we found that canonical *HTT* exon 1- exon 3 mRNA is more abundant than the aberrant exon 1-intron 1 RNA in heterozygous KI mice (Supplementary Fig. 8d, e). Using real time PCR to quantify the relative levels of exon 1-exon 3 and exon 1-intron 1, we verified that the aberrant spliced *HTT* mRNA (exon 1-intron 1) is specifically present in the KI mouse striatum but its level is lower than normally spliced mutant *HTT* (exon 1-exon 3) (Supplementary Fig. 8 f-g). The results suggest that, although the aberrant exon 1-intron 1 RNA can produce mutant exon 1 *HTT*, this small N-terminal *HTT* fragment may be predominantly generated from proteolytic cleavage of full-length or a longer *HTT* protein fragments."

In the discussion, we also included the following statement:

“Because mutant exon 1 HTT can be constantly generated by proteolytic cleavage and aberrant splicing of exon 1²⁰, genetic targeting should be focused on the production of exon 1 HTT.”

4. The authors’ conclusion that region-specific expression of HspBP1 is responsible for the accumulation of mutant Huntingtin could be strengthened by performing immunohistochemistry to complement the western data that is provided in the manuscript. In the HD knock-in models, mutant Huntingtin nuclear accumulation and aggregation also occurs in cortical layer V, in the hippocampus (of older mice), and in the piriform cortex. Detection of enhanced immunohistochemical staining of HspBP1 in these brain regions can provide complementary data strengthening the authors’ hypothesis.

We appreciate this important suggestion and performed additional immunohistochemical studies. The results indeed show that HspBP1 is enriched in the striatum when compared with other brain regions. We have provided this new data in Supplementary Fig. 11.

5. Alternatively, data mining of single-cell RNASeq or region-specific RNASeq databases may also provide additional confirmation provided that the cell and region-specific elevation of HspBP1 is under transcriptional control.

The reviewer is correct about the expression of mutant HTT aggregates in the brain regions in HD KI mice. However, the striatum shows the earliest accumulation of mutant HTT as compared with other brain regions. As mentioned above, the new immunohistochemical results indeed support the Western blotting result that HspBP1 is more abundant in the striatum (Supplementary Fig. 11). This new data complementarily supports the idea that the relatively high level of HspBP1 in the striatum could contribute to the preferential accumulation of mutant HTT in the mouse striatum.

We also performed RT-PCR analysis of HspBP1 in different brain regions of wild type mice but did not find that HspBP1 is regulated at the transcriptional level. The protein level of HspBP1 is not necessarily correlated with its mRNA level alteration (Sedlackova L et al., 2011). Thus, the brain regional difference in HspBP1 protein expression is more likely regulated at the protein level. We added this possibility when describing the expression of HspBP1 in the revision. However, this possibility would require an independent study to address.

Minor comments and questions:

1. Line 295: “...HTT is prone to misfold and that chaperone is important..” should be “...HTT is prone to misfold and that chaperones are important...” Line 297: “effects on chaperone and account” should be “effects on chaperone activity and account”

Thanks for reviewer's corrections and we have made these corrections in the revision.

2. Mouse behavioral studies: were both males and females used in the experiments?

We used both male and female mice for behavioral studies, which had been indicated in the method.

3. Legend to Fig. 5b: Please include the antibody used for this blot (mEM48?).

We have included mEM48 in the legend, which is Fig. 5a in the revision.

4. Legend to Fig. 7a: please include the antibody used for this blot. Fig. 7b: Are Hsp70 and CHIP levels also affected by aging?

We have added mEM48 to the Fig. 5a and 7a legends. As for Hsp70 and CHIP expression during aging, this would require an independent study. This is because there are a variety of antibodies to Hsp70 and CHIP, and previously published literature showed that Hsp70 is either slightly reduced or increased in aged rodent brains when different antibodies were used (Bodega G et al., 2002; Carnemolla A et al., 2014). Thus, to obtain a conclusive result, substantial experiments using aged mouse brains with different antibodies need to be done in an independent study.

Reviewer #2 (Remarks to the Author):

The document entitled 'Truncation of mutant huntingtin in knock-in mice via CRISPR-Cas9 uncovers exon 1 huntingtin as a key pathogenic form' by Yang et al. makes use of CRISPR-Cas9 to produce knock-in mouse models expressing various truncated versions of the mutant huntingtin protein. The nuclear aggregation and expression of exon 1 is compared between the knock-in lines of mice. These studies indicate the truncation of mHTT does not alter the production or localization of exon 1. Behaviour experiments and assessment of neuropathology further indicate that huntingtin of various lengths has the same degree of pathogenicity. Finally the authors show that nuclear aggregation can be prevented by inhibiting HspBP1.

General comments

1. Rationale for the study is interesting but the characterization of the behaviour and neuropathology of the mice is very superficial which prevents firm conclusions about the impacts of truncating mHTT

There are a number of HD KI mouse models, including Q175, Q150, Q111, Q91. These KI mice carry different repeats and were generated using different targeting approaches such that their genomic backgrounds and behavioral phenotypes are not identical. The one we used in our study is HD140Q KI model, which develops mild phenotypes and neuropathology, perhaps

because it expresses exon 1 of human HTT with 140Q at the endogenous level. The more robust phenotypes of HD140Q KI mice were found in homozygous KI mice. Thus, almost all published studies used homozygous HD140Q KI mice for non-motor behavioral analyses (Dorner et al., 2007; Hickey et al., 2008; Yu et al., 2017; Song et al., 2018; Franich et al., 2019).

In our studies, however, we had to use heterozygous HD140Q KI mice to compare with the newly established KI-96 and KI-571 KI mice because these new KI mice are heterozygous and homozygous KI mice are not viable. The behavioral phenotype of heterozygous HD140Q KI mice are obviously milder than other types of homozygous KI mice, but the motor function deficits could be reliably detected. For example, Rising et al found that accelerating rotarod performance deficits in heterozygous HD140Q KI mice became statistically significant at 11 months of age (Rising et al., 2011). Our preliminary studies did not reveal significant changes in non-motor behaviors of heterozygous HD140Q KI mice when they started to display defective rotarod and balance beam performance at the age of 8-9 months. Thus, we used motor function assays to compare all heterozygous KI mice of different genotypes in our studies.

In the revision, we added the rationale as the follows when describing the behavioral study:

“We examined motor functions of heterozygous HD KI mice to compare their phenotypes because homozygous KI-96 and KI-571 are not viable and the motor deficits in HD KI mice have been reliably replicated in many previous studies^{9,16,32,33}.”

Although heterozygous HD140Q KI mice showed mild neuropathologic changes, HTT aggregates and gliosis (reactive astrocytes) in HD KI mice have been repeatedly reported by different groups (Ishiguro et al., 2001; Lin et al., 2001; Palfi et al. 2007; Hickey et al., 2008; Yu et al., 2003; Hong et al., 2016; Agostoni et al., 2016). Thus, these pathological phenotypes allowed us to rigorously compare HD140Q KI (KI-FL) and KI-571 mice to assess their neuropathological changes. By comparing important and reliable phenotypes of different KI mouse model rather than comprehensive analyses of different phenotypes of these KI mice, we hope that we can present the most straightforward and key evidence to support the important role of N-terminal mutant HTT in HD pathogenesis.

We also added the above rationale in the revision as follows:

“Although heterozygous HD140Q KI mice showed mild neuropathologic changes, HTT aggregates and gliosis (increased Gfap) in HD KI mice have been repeatedly reported by different groups^{8,11,32,34-36}.”

2. The figures could use further polishing and contain many crocked lines and titles that are not centered

We thank the reviewer for pointing out this and have improved the figures in the revision.

3. Many figures are lacking quantifications that would be essential to support the claims stated in the text

We have included statistical data for those results that show differences in many figures in the previous version. Although reviewer-1 pointed out that statistical analyses are appropriate, and sufficient information is provided to allow reproduction of the work in other laboratories, we think that there is room for us to improve the qualification in this study. We added more statistical information and results to figures. These include new quantitative data in Fig. 2e; Fig. 3b; Fig. 5a; Fig. S8c, g, h; Fig. S12.

4. The story is difficult to follow and some of the figure panels are redundant

We have removed some redundancy in result presentation and also highlighted the differences in the presentations of each figure.

5. The N for biochemistry experiments is very low (N=3)

The N number normally indicated the animal numbers for biochemical and histological experiments, but the results were usually repeated multiple times for statistical analysis. We have clarified N numbers in figure legends.

Major concern

1. In Figure 1b for Exon-D the numbers of animals in the three groups does not add up to the total listed (6+7+12 =25 not 39).

This was a mistake, and we provided the updated information for this figure.

2. Figures 1 and 2 indicate that one of the studied proteins was produced by targeting Exon 13 (KI-571) but in figure 5 it is stated that KI-571 is produced by cleavage of exon 6.

We appreciate reviewer for pointing out this error. It should be exon 13, not exon 6, which should be consistent with Figure 1 and 2.

3. Explanations of the rationale for cleaving at Exons 2 and 6 are missing. Given that one of the purposes of this study is to understand if the various cleavage products of mHTT contribute to disease pathology it would be useful to have further details about how the various Knock-in lines impact the cleavage of mHTT.

As pointed out above, it should be cleaving at Exon 13, not Exon 6, to generate N-terminal HTT with about 571 amino acids. It has been reported that N-terminal HTT fragments with similar size are generated by caspase cleavage and are likely to contribute to HD pathogenesis. For example, N-terminal fragment containing D572 and D586 was generated by caspase-1 (Martin et al., 2019) and caspase-6 (Graham et al., 2006), respectively. Caspase-6-

cleaved product was extensively investigated by different groups but its pathological role was demonstrated by inconsistent results (Tebbenkamp et al., 2011; Landles et al., 2012; Gafni et al., 2012; Ehrnhoefer et al., 2019). Thus, our studies were focused on exon 1 HTT and N-terminal HTT (571). We have emphasized these to strengthen our rationale for targeting these two regions as follows:

“To further test this hypothesis, we used stereotaxic injection of AAV CRISPR/Cas9 to truncate the *HTT* gene at the region corresponding to amino acid 91 and 571 in adult KI-FL mouse brain. These N-terminal HTT fragments are similar to exon 1 HTT and caspase-cleaved products^{25,26}, respectively, which have been extensively investigated for their roles in HD pathogenesis²⁷⁻³⁰.”

In the discussion, we also included following discussion:

“Previous studies have found that caspase-6 cleavage at 586 aa generates N-terminal HTT fragments²⁵. However, there are conflicting results about the beneficial effects of blocking caspase-6 cleavage in mutant HTT^{28,29}. Because full-length HTT is constantly processed by proteolysis cleavage, it is likely that only those proteins that are able to interact with exon 1 HTT may stably associate with mutant HTT and may be involved in HD pathogenesis. Indeed, many studies have found that posttranslational modulations in exon 1 HTT or altering the first 17 amino acids of HTT can significantly regulate HTT toxicity⁵⁵⁻⁵⁷.”

4. In Figure 2b and d western blots of KI-571 and KI-96 are shown which indicate that these mice do not have full length mHTT but they do express exon 1 mHTT. I find it odd that a longer molecular weight band is not detected. The authors should discuss this.

This is an interesting and important issue that was investigated in our studies. We also thought that KI-571 mice should generate longer HTT fragments than KI-96 mice. However, we were unable to identify them. Our studies suggest that a small N-terminal mutant HTT fragment (exon 1 HTT) is more stable than other fragments. Also, aberrant splicing of exon 1 could also contribute to the production of exon 1 HTT. These possibilities are supported by the following evidence: (1) all KI mice have this stable fragment; (2) all KI mice show the preferential nuclear distribution and aggregation of N-terminal mutant HTT; and (3) all KI mice display similar phenotypes and pathological changes. Thus, our studies demonstrate that exon 1 HTT is the major pathological form when mutant HTT is expressed at the endogenous level. As indicated by reviewer-1, our findings will be of interest to both the HD community and to a wider audience, as they provide new data supporting the importance of the mutant Huntingtin exon 1 fragment in HD pathogenesis.

We have included the following discussion in the revision:

“Different HTT fragments may be continuously cleaved to form the exon 1 HTT product. In the HD brains, mutant exon 1 HTT is more stable because a large polyQ repeat may cause its misfolding and aggregation that is resistant to further digestion.”

5. Figures 2e, 3b-c and 4a all show aggregation in the striatum and are redundant

To conclude that all KI mice have the similar patterns of nuclear HTT accumulation and aggregation, it is important to compare HTT aggregates in different KI mouse models at different ages and in different brain regions. Fig. 2e is to compare three KI mouse lines for nuclear HTT in the striatum at 6 and 11 months. Fig. 3B-C are to compare HTT aggregates in different brain regions. We revised the figures by removing some redundant images and emphasized statistical results of comparing different brain regions for the presence of nuclear HTT. Fig. 4a is to compare R6/2 with all KI mice for HTT aggregation. This comparison is necessary, as it indicates that the expression level of mutant exon 1 HTT is very important for severe neuropathology and that all KI mice show the lower level of mutant exon 1 HTT and much less abundant HTT aggregates than R6/2 mice so that their phenotypes are much milder than R6/2 mice.

6. In the results section it is stated the KI-96 has a worse score on the balance beam than the other knock-in lines but this is not indicated with statistics on the graph and is inconsistent with what is shown at later time points where this model seems to have improved performance. This data should be discussed further.

Figure 6A actually shows that KI-96 mice did have a poor score on the balance beam starting at 9 months and that this performance was even worse at 11 months as compared with 9 months, as it took longer time for KI-96 mice to cross the beam at 11 months of age. It is known that the motor function deficits of KI mice are age-dependent.

7. To conclude that the truncation of mHTT does not impact neuropathology and behaviour a much broader range of assessments would be required. Astrogliosis is far from the most apparent or important neuropathological hallmark of disease and cognitive and non-motor performance are also important aspects of pathology.

These are the same issue as the first issue. Please see the responses above to the first issue.

8. While the data on HspBP1 is interesting it seems rather disjointed from the main focus of the paper.

It would be interesting to know why all KI mice display the preferential nuclear distribution of mutant HTT in the neurons in the striatum. We think that HspBP1 is a good candidate for this preferential localization. As pointed out by reviewer-1, the authors now also provide evidence suggesting that the abundant striatal expression of HspBP1 in the HD model mice is responsible for the preferential accumulation of mutant Huntingtin in the striatum. Thus, including HspBP1 results provides some insight into the important phenomenon of the striatal accumulation of mutant HTT. To this end, we included new immunohistochemical results showing the more abundant level of HspBP1 protein in the mouse striatum (new Supplementary Fig. 11).

Minor concerns

1. For Figure 6 a two-way ANOVA was performed to assess all behaviour data. However a repeated measures ANOVA would be more appropriate if the same animals were used at multiple time points.

For Fig 6, we started to use more than 14 mice per genotype for behavioral studies. We used a two-way ANOVA to analyze, but it would be hard to perform repeated measures ANOVA because during this time course study from 6 to 11 months, some mice were used for isolating their brains for biochemical and immunocytochemical studies.

2. In the results section titled Preferential nuclear accumulation of exon 1 like HTT in the striatum many of the references to figures seem to be inaccurate (Lines 169, 170, 182 and 184)

We have corrected the references to figures.

3. In Figure 2e data is not shown for KI-96 at 11 m. Given that these mice appear to have a different behavioural phenotype it would be important to see the nuclear aggregation at later ages.

We added KI-96 data at 11 m to this figure and also included quantitative results.

References

Agostoni E, Michelazzi S, Maurutto M, Carnemolla A, Ciani Y, Vatta P, Roncaglia P, Zucchelli S, Leanza G, Mantovani F, Gustincich S, Santoro C, Piazza S, Del Sal G, Persichetti F. Effects of Pin1 Loss in Hdh(Q111) Knock-in Mice. *Front Cell Neurosci.* 2016 May 2;10:110.

Bodega G, Hernández C, Suárez I, Martín M, Fernández B. HSP70 constitutive expression in rat central nervous system from postnatal development to maturity. *J Histochem Cytochem.* 2002;50(9):1161–1168.

Carnemolla A, Labbadia JP, Lazell H, Neueder A, Moussaoui S, Bates GP. Contesting the dogma of an age-related heat shock response impairment: implications for cardiac-specific age-related disorders. *Hum Mol Genet.* 2014;23(14):3641–3656.

Cornett J, Cao F, Wang CE, Ross CA, Bates GP, Li SH, Li XJ. Polyglutamine expansion of huntingtin impairs its nuclear export. *Nat Genet.* 2005 Feb;37(2):198-204.

Davies SW, Turmaine M, Cozens BA, DiFiglia M, Sharp AH, Ross CA, Scherzinger E, Wanker EE, Mangiarini L, Bates GP. Formation of neuronal intranuclear inclusions underlies the neurological dysfunction in mice transgenic for the HD mutation. *Cell.* 1997 Aug 8;90(3):537-48.

Dorner JL, Miller BR, Barton SJ, Brock TJ, Rebec GV. Sex differences in behavior and striatal ascorbate release in the 140 CAG knock-in mouse model of Huntington's disease. *Behav Brain Res.* 2007 Mar 12;178(1):90-7.

Ehrnhoefer DE, Skotte NH, Reinshagen J, Qiu X, Windshügel B, Jaishankar P, Ladha S, Petina O, Khankischpur M, Nguyen YTN, Caron NS, Razeto A, Meyer Zu Rheda M, Deng Y, Huynh KT, Wittig I, Gribbon P, Renslo AR, Geffken D, Gul S, Hayden MR. Activation of Caspase-6 Is Promoted by a Mutant Huntingtin Fragment and Blocked by an Allosteric Inhibitor Compound. *Cell Chem Biol.* 2019 Sep 19;26(9):1295-1305.

Franich NR, Hickey MA, Zhu C, Osborne GF, Ali N, Chu T, Bove NH, Lemesre V, Lerner RP, Zeitlin SO, Howland D, Neueder A, Landles C, Bates GP, Chesselet MF. Phenotype onset in Huntington's disease knock-in mice is correlated with the incomplete splicing of the mutant huntingtin gene. *J Neurosci Res.* 2019 Dec;97(12):1590-1605.

Gafni J, Papanikolaou T, Degiacomo F, Holcomb J, Chen S, Menalled L, Kudwa A, Fitzpatrick J, Miller S, Ramboz S, Tuunanen PI, Lehtimäki KK, Yang XW, Park L, Kwak S, Howland D, Park H, Ellerby LM. Caspase-6 activity in a BACHD mouse modulates steady-state levels of mutant huntingtin protein but is not necessary for production of a 586 amino acid proteolytic fragment. *J Neurosci.* 2012 May 30;32(22):7454-65.

Graham RK, Deng Y, Slow EJ, Haigh B, Bissada N, Lu G, Pearson J, Shehadeh J, Bertram L, Murphy Z, Warby SC, Doty CN, Roy S, Wellington CL, Leavitt BR, Raymond LA, Nicholson DW, Hayden MR. Cleavage at the caspase-6 site is required for neuronal dysfunction and degeneration due to mutant huntingtin. *Cell.* 2006 Jun 16;125(6):1179-91.

Hickey MA, Kosmalska A, Enayati J, Cohen R, Zeitlin S, Levine MS, Chesselet MF. Extensive early motor and non-motor behavioral deficits are followed by striatal neuronal loss in knock-in Huntington's disease mice. *Neuroscience.* 2008 Nov 11;157(1):280-95.

Hong Y, Zhao T, Li XJ, Li S. Mutant Huntingtin Impairs BDNF Release from Astrocytes by Disrupting Conversion of Rab3a-GTP into Rab3a-GDP. *J Neurosci.* 2016 Aug 24;36(34):8790-801.

Ishiguro H, Yamada K, Sawada H, Nishii K, Ichino N, Sawada M, Kurosawa Y, Matsushita N, Kobayashi K, Goto J, Hashida H, Masuda N, Kanazawa I, Nagatsu T. Age-dependent and tissue-specific CAG repeat instability occurs in mouse knock-in for a mutant Huntington's disease gene. *J Neurosci Res.* 2001 Aug 15;65(4):289-97.

Landles C, Sathasivam K, Weiss A, Woodman B, Moffitt H, Finkbeiner S, Sun B, Gafni J, Ellerby LM, Trotter Y, Richards WG, Osmand A, Paganetti P, Bates GP. Proteolysis of mutant huntingtin produces an exon 1 fragment that accumulates as an aggregated protein in neuronal nuclei in Huntington disease. *J Biol Chem.* 2010 Mar 19;285(12):8808-23.

Landles C, Weiss A, Franklin S, Howland D, Bates G. Caspase-6 does not contribute to the proteolysis of mutant huntingtin in the HdhQ150 knock-in mouse model of Huntington's disease. *PLoS Curr.* 2012 Jul 16;4:e4fd085bfc9973.

Sedlackova L, Sosna A, Vavrincova P, et al. Heat shock protein gene expression profile may differentiate between rheumatoid arthritis, osteoarthritis, and healthy controls. *Scand J Rheumatol*. 2011;40(5):354–357

Lin CH, Tallaksen-Greene S, Chien WM, Cearley JA, Jackson WS, Crouse AB, Ren S, Li XJ, Albin RL, Detloff PJ. Neurological abnormalities in a knock-in mouse model of Huntington's disease. *Hum Mol Genet*. 2001 Jan 15;10(2):137-44.

Martin DDO, Schmidt ME, Nguyen YT, Lazic N, Hayden MR. Identification of a novel caspase cleavage site in huntingtin that regulates mutant huntingtin clearance. *FASEB J*. 2019 Mar;33(3):3190-3197.

Palfi S, Brouillet E, Jarraya B, Bloch J, Jan C, Shin M, Condé F, Li XJ, Aebischer P, Hantraye P, Déglon N. Expression of mutated huntingtin fragment in the putamen is sufficient to produce abnormal movement in non-human primates. *Mol Ther*. 2007 Aug;15(8):1444-51.

Rising AC, Xu J, Carlson A, Napoli VV, Denovan-Wright EM, Mandel RJ. Longitudinal behavioral, cross-sectional transcriptional and histopathological characterization of a knock-in mouse model of Huntington's disease with 140 CAG repeats. *Exp Neurol*. 2011 Apr;228(2):173-82.

Song H, Li H, Guo S, Pan Y, Fu Y, Zhou Z, Li Z, Wen X, Sun X, He B, Gu H, Zhao Q, Wang C, An P, Luo S, Hu Y, Xie X, Lu B. Targeting Gpr52 lowers mutant HTT levels and rescues Huntington's disease-associated phenotypes. *Brain*. 2018 Jun 1;141(6):1782-1798.

Tebbenkamp AT, Green C, Xu G, Denovan-Wright EM, Rising AC, Fromholt SE, Brown HH, Swing D, Mandel RJ, Tessarollo L, Borchelt DR. Transgenic mice expressing caspase-6-derived N-terminal fragments of mutant huntingtin develop neurologic abnormalities with predominant cytoplasmic inclusion pathology composed largely of a smaller proteolytic derivative. *Hum Mol Genet*. 2011 Jul 15;20(14):2770-82.

Wang CE, Zhou H, McGuire JR, Cerullo V, Lee B, Li SH, Li XJ. Suppression of neuropil aggregates and neurological symptoms by an intracellular antibody implicates the cytoplasmic toxicity of mutant huntingtin. *J Cell Biol*. 2008 Jun 2;181(5):803-16.

Yang S, Yang H, Huang L, Chen L, Qin Z, Li S, Li XJ. Lack of RAN-mediated toxicity in Huntington's disease knock-in mice. *Proc Natl Acad Sci U S A*. 2020 Feb 6:201919197. doi: 10.1073/pnas.1919197117. Epub ahead of print. PMID: 32029588..

Yu M, Fu Y, Liang Y, Song H, Yao Y, Wu P, Yao Y, Pan Y, Wen X, Ma L, Hexige S, Ding Y, Luo S, Lu B. Suppression of MAPK11 or HIPK3 reduces mutant Huntingtin levels in Huntington's disease models. *Cell Res*. 2017 Dec;27(12):1441-1465.

Yu ZX, Li SH, Evans J, Pillarisetti A, Li H, Li XJ. Mutant huntingtin causes context-dependent neurodegeneration in mice with Huntington's disease. *J Neurosci*. 2003 Mar 15;23(6):2193-202.

Zhou H, Cao F, Wang Z, Yu ZX, Nguyen HP, Evans J, Li SH, Li XJ. Huntingtin forms toxic NH₂-terminal fragment complexes that are promoted by the age-dependent decrease in proteasome activity. *J Cell Biol.* 2003 Oct 13;163(1):109-18.

Reviewer #1 (Remarks to the Author):

The authors have made a very good effort to address the comments and issues that were raised about the original version of their manuscript. In particular, to further evaluate how much aberrant Htt exon1-intron levels contribute to the total levels of the mHTT exon 1 fragment, the authors now provide immunostaining of the KI-571 and KI-FL striatum with 1C2 and mEM48 antibodies (Supplemental Fig. 6), and improved western blots in Supplemental Fig. 5. In addition, the authors have performed RNASeq to characterize the expression of the incompletely spliced form of mHtt transcript and performed additional RT-PCRs with appropriate controls (Supplementary Fig. 8). To strengthen the conclusion that region-specific expression of HspBP1 is responsible for the accumulation of mutant HTT fragments in the striatum, the authors have performed additional immunohistochemical experiments demonstrating that HspBP1 is enriched in the striatum (Supplementary Fig. 11). Taken altogether, the authors' new data, and edits to the text have addressed my concerns and substantially improved their manuscript.

Point-by-point response

REVIEWERS' COMMENTS:

Reviewer #1 (Remarks to the Author):

The authors have made a very good effort to address the comments and issues that were raised about the original version of their manuscript. In particular, to further evaluate how much aberrant Htt exon1-intron levels contribute to the total levels of the mHTT exon 1 fragment, the authors now provide immunostaining of the KI-571 and KI-FL striatum with 1C2 and mEM48 antibodies (Supplemental Fig. 6), and improved western blots in Supplemental Fig. 5. In addition, the authors have performed RNASeq to characterize the expression of the incompletely spliced form of mHtt transcript and performed additional RT-PCRs with appropriate controls (Supplementary Fig. 8). To strengthen the conclusion that region-specific expression of HspBP1 is responsible for the accumulation of mutant HTT fragments in the striatum, the authors have performed additional immunohistochemical experiments demonstrating that HspBP1 is enriched in the striatum (Supplementary Fig. 11). Taken altogether, the authors' new data, and edits to the text have addressed my concerns and substantially improved their manuscript.

Thanks for the reviewer's comments on our revision!

Xiao-Jiang Li